# Understanding Representation Dynamics of Diffusion Models via Low-Dimensional Modeling

**Xiao Li**[*]
University of Michigan
xlxiao@umich.edu

**Zekai Zhang**[*]
University of Michigan
zzekai@umich.edu

**Xiang Li**
University of Michigan
forkobe@umich.edu

**Siyi Chen**
University of Michigan
siyich@umich.edu

**Zhihui Zhu**
Ohio State University
zhu.3440@osu.edu

**Peng Wang**
University of Michigan
pengwa@umich.edu

**Qing Qu**
University of Michigan
qingqu@umich.edu

## Abstract

Diffusion models, though originally designed for generative tasks, have demonstrated impressive self-supervised representation learning capabilities. A particularly intriguing phenomenon in these models is the emergence of unimodal representation dynamics, where the quality of learned features peaks at an intermediate noise level. In this work, we conduct a comprehensive theoretical and empirical investigation of this phenomenon. Leveraging the inherent low-dimensionality structure of image data, we theoretically demonstrate that the unimodal dynamic emerges when the diffusion model successfully captures the underlying data distribution. The unimodality arises from an interplay between denoising strength and class confidence across noise scales. Empirically, we further show that, in classification tasks, the presence of unimodal dynamics reliably reflects the diffusion model's generalization: it emerges when the model generate novel images and gradually transitions to a monotonically decreasing curve as the model begins to memorize the training data.

## 1 Introduction

Diffusion models, a class of score-based generative models, have achieved great empirical success in various tasks such as image and video generation, speech and audio synthesis, and solving inverse problems [1–15]. In general, these models, consisting of forward and backward processes, learn data distributions by simulating the non-equilibrium thermodynamic diffusion process [2, 16, 17]. Specifically, the forward process progressively adds Gaussian noise to training samples until they are fully corrupted, while the backward process involves learning a score-based model to generate samples from the noisy inputs [17, 18].

Beyond their strong generative capabilities, recent studies have revealed that diffusion models also possess remarkable representation learning capabilities [19–24]. In particular, the internal feature extractors of trained diffusion models have been shown to serve as powerful self-supervised learners, achieving strong performance on downstream tasks such as classification [20, 21], semantic

---

[*]The first two authors contributed equally to the work.

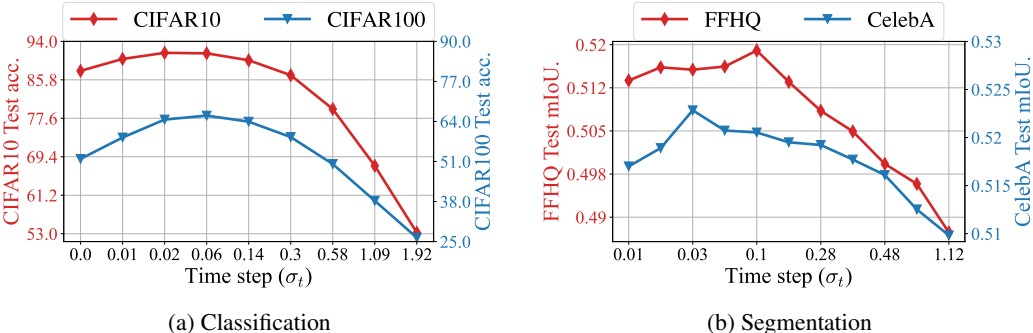

(a) Classification        (b) Segmentation

Figure 1: **Unimodal representation dynamics in diffusion-based representation learning tasks.** This unimodal representation pattern has been previously observed in diffusion-based representation learning tasks; see [19, 20, 23]. To verify this, we train diffusion models on various datasets and evaluate downstream performance using noisy images $x_t$ at different timesteps $t$. In both classification and segmentation tasks, the performance consistently follows a unimodal trend, peaking at intermediate noise levels. In (b), "mIoU" denotes mean Intersection over Union, a standard metric used in segmentation tasks.

segmentation [19], and image alignment [23]. In many cases, these representations match or even surpass those from specialized self-supervised methods, implying that diffusion models have the potential to unify generative and discriminative learning in vision.

Alongside the empirical success, a prevalent phenomenon, which we refer to as the *unimodal representation dynamics*, has been widely observed in diffusion models for representation learning [19, 20, 23]: the quality of learned representations, as measured by downstream task performance, follows a unimodal trend across noise levels. Specifically, the most effective features consistently appear at an intermediate noise level, while performance degrades as inputs become either fully noisy or entirely clean (see Figure 1). Despite being widely observed, the underlying cause of this unimodal representation dynamic remains poorly understood.

**Our contributions.** In this work, we investigate the emergence of unimodal representation dynamics in diffusion models and its cause. Our analysis characterizes how representation quality varies across noise levels and offers new perspectives on how diffusion models, despite their generative design, can excel at representation learning. Specifically, we conduct a comprehensive theoretical and empirical study of unimodal representation dynamics in diffusion models. Motivated by the well-established observation that natural image data typically reside on a low-dimensional manifold [25–27], we examine the representation dynamics through how diffusion models learn from noisy mixtures of low-rank Gaussian (`MoLRG`) distributions. We show that the emergence of unimodal representations is intrinsically linked to the model's ability to capture the underlying low-dimensional structure of the data. Our key contributions are as follows:

- **Mathematical framework for studying representation learning in diffusion models.** To analyze the representation dynamics, we provide a mathematical study of how diffusion models learn from `MoLRG` distributions, where data lie on a union of low-dimensional subspaces. We adopt a simplified yet analytically tractable network architecture that mimics key structural properties of U-Net. Under this setting, we quantify the quality of learned representations via the Signal-to-Noise Ratio (SNR) within the target subspace, enabling us to characterize how representation quality evolves across timesteps in the diffusion process.

- **Theoretical explanation for the emergence of unimodal dynamics.** Leveraging the structures of the `MoLRG` model, we prove that the unimodal pattern in representation quality naturally arises when the model effectively captures the low-dimensional data distribution. We show that this unimodal pattern is driven by an interplay between denoising strength and class separability that varies with noise levels. There exists an intermediate diffusion timestep where class-irrelevant components are maximally suppressed and class-relevant features are best preserved, resulting in optimal representation quality.

- **Empirical connection between unimodal dynamics and generalization.** Empirically, we demonstrate that the presence of unimodal representation dynamics serves as a reliable indicator of model generalization in classification tasks. Specifically, the unimodal pattern consistently emerges when the model generalizes well, but progressively vanishes as the model shifts toward memorizing the training data.

**Relationship to prior results.** Recent empirical advances in leveraging diffusion models for downstream representation learning have gained significant attention. However, a theoretical understanding of how diffusion models learn representations across different noise levels remains largely unexplored. Here, we focus on the results most relevant to our work and defer a more comprehensive survey to Appendix A. A recent study by [24] takes initial steps in this direction by analyzing the optimization dynamics of a two-layer CNN trained with diffusion loss on binary class data. Their focus is on contrasting the learning behavior under denoising versus classification objectives, without examining how representation quality evolves across timesteps. In contrast, our work characterizes and compares representations learned at different timesteps, provides a deeper understanding of diffusion-based representation learning and also extends to multi-class settings. A recent study by [28] also investigates the influence of timesteps in diffusion-based representation learning, focusing on attribute classification and counterfactual generation. In contrast, our work provides a theoretical explanation and practical metric for the emergence of unimodal representation dynamics and shows its relationship with data and model complexity and training iterations.

## 2 Problem setup

**Basics of diffusion models.** Diffusion models are a class of probabilistic generative models that aim to reverse a progressive noising process by mapping an underlying data distribution $p_{\text{data}}$ to a Gaussian distribution. The forward diffusion process begins with clean data $\boldsymbol{x}_0 \sim p_{\text{data}}$ and adds Gaussian noise over time $t \in [0, 1]$, which can be described by the following stochastic differential equation (SDE):

$$\mathrm{d}\boldsymbol{x}_t = f(t)\boldsymbol{x}_t\,\mathrm{d}t + g(t)\,\mathrm{d}\boldsymbol{w}_t,$$

where $f(t)$ is the drift coefficient, $g(t)$ is the diffusion coefficient, and $\{\boldsymbol{w}_t\}$ is a standard Wiener process. Then, one can verify that the noisy data satisfy $\boldsymbol{x}_t = s_t\boldsymbol{x}_0 + s_t\sigma_t\boldsymbol{\epsilon}$ with $\boldsymbol{\epsilon} \sim \mathcal{N}(\mathbf{0}, \boldsymbol{I})$, where $s_t$ and $\sigma_t$ are scaling factors determined by $f(t)$ and $g(t)$. For ease of exposition, let $p_t(\boldsymbol{x})$ denote the probability density function (pdf) of the noisy data $\boldsymbol{x}_t$ for each $t \in [0, 1]$. In particular, $p_0 = p_{\text{data}}$. To simplify the analysis, we assume throughout the paper that $s_t = 1$.

The reverse process transforms noise back into clean data by leveraging the score function $\nabla \log p_t(\boldsymbol{x}_t)$ and is governed by the reverse SDE [29]:

$$\mathrm{d}\boldsymbol{x}_t = \big(f(t)\boldsymbol{x}_t - g^2(t)\nabla \log p_t(\boldsymbol{x}_t)\big)\,\mathrm{d}t + g(t)\mathrm{d}\bar{\boldsymbol{w}}_t,$$

where $\{\bar{\boldsymbol{w}}_t\}$ is a Wiener process independent of $\{\boldsymbol{w}_t\}$. This enables diffusion models to generate new samples from the underlying data distribution $p_{\text{data}}$ by initializing from pure Gaussian noise and iteratively denoising via the score function.

**Training loss of diffusion models.** Note that the score function $\nabla \log p_t(\boldsymbol{x}_t)$ depends on the unknown data distribution $p_{\text{data}}$. According to Tweedie's formula [30], i.e.,

$$\mathbb{E}\left[\boldsymbol{x}_0 | \boldsymbol{x}_t\right] = \boldsymbol{x}_t + \sigma_t^2 \nabla \log p_t(\boldsymbol{x}_t), \tag{1}$$

we can alternatively estimate $\nabla \log p_t(\boldsymbol{x}_t)$ by training a network $\boldsymbol{x}_{\boldsymbol{\theta}}(\boldsymbol{x}_t, t)$, which is referred to as denoising autoencoder (DAE), to approximate the posterior mean $\mathbb{E}[\boldsymbol{x}_0 | \boldsymbol{x}_t]$ [20, 22, 31]. To learn the network parameter $\boldsymbol{\theta}$, we minimize the following empirical loss:

$$\min_{\boldsymbol{\theta}}\ \mathcal{L}(\boldsymbol{\theta}) := \sum_{i=1}^{N} \int_0^1 \lambda_t\, \mathbb{E}_{\boldsymbol{\epsilon}\sim\mathcal{N}(\mathbf{0}, \boldsymbol{I}_n)} \left[\left\|\boldsymbol{x}_{\boldsymbol{\theta}}\left(\boldsymbol{x}_0^{(i)} + \sigma_t\boldsymbol{\epsilon}, t\right) - \boldsymbol{x}_0^{(i)}\right\|^2\right]\,\mathrm{d}t, \tag{2}$$

where $\boldsymbol{x}_0^{(i)} \overset{i.i.d.}{\sim} p_{\text{data}}$ for all $i = 1, \ldots, N$ denote the training samples and $\lambda_t$ represents the weight associated with each noise level.

# 3 Good distribution learning implies unimodal representation dynamics

Building on the setup introduced in Section 2, this section provides a mathematical framework for analyzing the representation quality of diffusion models, and theoretically we characterize the unimodal representation dynamics exhibited by diffusion models across noise levels. We validate our theoretical findings on both synthetic and real-world datasets.

## 3.1 Assumptions of data distribution

In this work, we assume that the underlying data distribution is *low-dimensional*, which follows a noisy version of the mixture of low-rank Gaussians (MoLRG) [32–34], defined as follows.

**Assumption 1** ($K$-Class Noisy MoLRG Distribution). *For any sample $x_0$ drawn from the noisy MoLRG distribution with $K$ subspaces, we have*

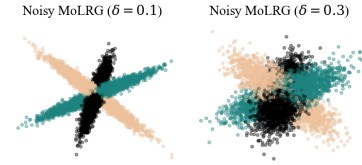

Figure 2: **An illustration of MoLRG with different noise levels.** We visualize samples drawn from noisy MoLRG with noise levels $\delta = 0.1$, $0.3$ and $K = 3$.

$$x_0 = U_k^\star a + \delta \widetilde{U}_k^\star e, \text{ with probability } \pi_k \geq 0, \quad (3)$$

*where (i) $U_k^\star \in \mathcal{O}^{n \times d_k}$ denotes an orthonormal basis of the $k$-th subspace for each $k = 1, \ldots, K$, $\widetilde{U}_k^\star \in \mathcal{O}^{n \times D_k}$ is an orthonormal basis for the $D_k$-dimensional subspace spanned by $\{U_l^\star : l \neq k\}$; (ii) $a \overset{i.i.d.}{\sim} \mathcal{N}(0, I_{d_k})$, $e \overset{i.i.d.}{\sim} \mathcal{N}(0, I_{D_k})$ are independent standard Gaussian vectors representing the latent signal and noise components of $x_0$, respectively; (iii) $\delta > 0$ controls the data noise level, and (iv) $\pi_k \geq 0$ is the mixing proportion satisfying $\sum_{k=1}^K \pi_k = 1$.*

One can verify that any sample $x_0$ drawn from the above distribution satisfies:

$$x_0 \sim \sum_{k=1}^K \pi_k \mathcal{N}\left(0, U_k^\star U_k^{\star\top} + \delta^2 \widetilde{U}_k^\star \widetilde{U}_k^{\star\top}\right)$$

As shown in Figure 2, data from MoLRG resides on a union of low-dimensional subspaces, each following a Gaussian distribution with a low-rank covariance matrix representing its basis perturbed by some noise. For simplicity, we assume equal subspace dimensions ($d_1 = \cdots = d_K = d$), orthogonal bases (i.e., $U_k^{\star\top} U_l^\star = 0$ for $k \neq l$), and uniform mixing weights ($\pi_1 = \cdots = \pi_K = 1/K$).[2] Additionally, the study of noisy MoLRG distributions is further motivated by the following facts:

- *MoLRG captures the intrinsic low-dimensionality of image data.* Although real-world image datasets are high-dimensional in terms of pixel count and data volume, extensive empirical studies [25–27] demonstrated that their intrinsic dimensionality is considerably lower. Additionally, recent work [35, 36] has leveraged the intrinsic low-dimensional structure of real-world data to analyze the convergence guarantees of diffusion model sampling. The MoLRG distribution, which models data in a low-dimensional space with rank $d_k \ll n$, effectively captures this property.

- *Latent diffusion models encourage the latent space toward a Gaussian distribution.* State-of-the-art large-scale diffusion models [37, 38] typically employ autoencoders [39] to project images into a latent space, where a KL penalty encourages the learned latent distribution to approximate standard Gaussians [3]. Furthermore, recent studies [22, 40] show that diffusion models can be trained to leverage the intrinsic subspace structure of real-world data.

- *Modeling the complexity of real-world image datasets.* The noise term $\delta \widetilde{U}_k^\star e_i$ captures perturbations outside the $k$-th subspace via the noise space $\widetilde{U}_k^\star$, analogous to insignificant attributes of real-world images, such as the background of an image. While additional noise may be less significant for representation learning, it plays a crucial role in enhancing the fidelity of generated samples.

---

[2]The assumptions of subspace orthogonality, equal subspace rank and uniform mixing weights are adopted for analytical simplicity. Empirical results in Appendix C demonstrate that the unimodal dynamics persist when these assumptions are relaxed.

In other words, our data model assumes that the overall dataset lies in a union of low-dimensional subspaces, where each class is associated with a dominant class-specific subspace, and there exists a shared subspace common across all classes that captures class-independent fine-grained details. This shared–specific decomposition has been empirically supported in the context of subspace clustering and representation learning [41, 42].

## 3.2 Network parametrization

To analyze how feature representations evolve across noise levels in diffusion models, we study the following nonlinear network parameterization inspired from the MoLRG data assumption.. Specifically, we parameterize the DAE $\boldsymbol{x_\theta}(\boldsymbol{x}_t, t)$ and its corresponding latent feature representation $\boldsymbol{h_\theta}(\boldsymbol{x}_t, t)$ as follows:

$$\boldsymbol{x_\theta}(\boldsymbol{x}_t, t) = \boldsymbol{U}\boldsymbol{h_\theta}(\boldsymbol{x}_t, t), \ \boldsymbol{h_\theta}(\boldsymbol{x}_t, t) = \boldsymbol{D}(\boldsymbol{x}_t, t)\boldsymbol{U}^\top \boldsymbol{x}_t, \ \boldsymbol{D}(\boldsymbol{x}_t, t) = \mathrm{diag}\left(\beta_1^t \boldsymbol{I}_d, \ldots, \beta_K^t \boldsymbol{I}_d\right), \quad (4)$$

where $\boldsymbol{\theta} = \{\boldsymbol{U}\}$ denotes a set of learnable parameters with $\boldsymbol{U} = [\boldsymbol{U}_1, \cdots, \boldsymbol{U}_K] \in \mathcal{O}^{n \times Kd}$. Let $\zeta_t = \frac{1}{1+\sigma_t^2}$ and $\xi_t = \frac{\delta^2}{\delta^2+\sigma_t^2}$, where $\sigma_t$ is the noise scaling in (1). Correspondingly, we parameterize $\beta_l^t = \xi_t + (\zeta_t - \xi_t)w_l(\boldsymbol{x}_t, t)$ in (4) with

$$w_l(\boldsymbol{x}_t, t) = \frac{\exp\left(g_l(\boldsymbol{x}_t, t)\right)}{\sum_{s=1}^K \exp\left(g_s(\boldsymbol{x}_t, t)\right)}, \quad g_l(\boldsymbol{x}_t, t) = \frac{\zeta_t}{2\sigma_t^2}\|\boldsymbol{U}_l^\top \boldsymbol{x}_t\|^2 + \frac{\xi_t}{2\sigma_t^2}\|\widetilde{\boldsymbol{U}}_l^\top \boldsymbol{x}_t\|^2. \quad (5)$$

This network architecture can be interpreted as a shallow U-Net [43] composed with a blockwise mixture-of-experts [44] mechanism, or equivalently, a restricted form of self-attention, with the following components:

- *Low-dimensional projection.* The input $\boldsymbol{x}_t$ is projected into a latent space via a learned orthonormal basis $\boldsymbol{U}^\top \boldsymbol{x}_t$, which is partitioned into $K$ groups of dimension $d$. Each block $\boldsymbol{U}_l$ can be viewed as an individual expert operating on a distinct subspace of the input.

- *Expert weighting.* Each projected latent group is then reweighted by a coefficient $\beta_l^t$, which depends on the input $\boldsymbol{x}_t$ and timestep $t$ via a softmax function $w_l(\boldsymbol{x}_t, t)$. These coefficients form the block-diagonal matrix $\boldsymbol{D}(\boldsymbol{x}_t, t)$, which scales each group independently. This structure can be interpreted as a mixture-of-experts model [44], where the contribution of each expert is modulated by the input. It may also be viewed as a restricted self-attention mechanism [45], where attention is applied at the group level rather than individually, yielding the feature representations of the input as in (4).

- *Symmetric reconstruction.* The modulated feature representation is projected back to the input space via the same expert blocks $\boldsymbol{U}$, forming a symmetric encoder–decoder architecture.

Moreover, this parameterization in (4) induces a time and data-dependent feature representation $\boldsymbol{h_\theta}(\boldsymbol{x}_t, t)$, which enables systematic analysis of representation quality across noise scales.

## 3.3 A metric for measuring representation quality

To understand diffusion-based representation learning under the MoLRG data model, we define the following signal-to-noise ratio (SNR) to measure the representation quality as follows.

**Definition 1.** Suppose the data $\boldsymbol{x}_0$ follows the noisy MoLRG introduced in Assumption 1. Without loss of generality, let $k \in [K]$ denote the true class of $\boldsymbol{x}_0$. For any trained DAE $\hat{\boldsymbol{x}}_{\boldsymbol{\theta}}$ parameterized as in (4), we define:

$$\mathrm{SNR}(\hat{\boldsymbol{x}}_{\boldsymbol{\theta}}, t) := \mathbb{E}_k\left[\frac{\mathbb{E}_{\boldsymbol{x}_t}[\|\boldsymbol{U}_k^\star \hat{\boldsymbol{h}}_{\boldsymbol{\theta}}(\boldsymbol{x}_t, t)\|^2| \ k]}{\mathbb{E}_{\boldsymbol{x}_t}[\|\hat{\boldsymbol{x}}_{\boldsymbol{\theta}}(\boldsymbol{x}_t, t) - \boldsymbol{U}_k^\star \hat{\boldsymbol{h}}_{\boldsymbol{\theta}}(\boldsymbol{x}_t, t)\|^2| \ k]}\right]. \quad (6)$$

where $\hat{\boldsymbol{x}}_{\boldsymbol{\theta}}(\boldsymbol{x}_t, t) = \boldsymbol{U}\,\hat{\boldsymbol{h}}_{\boldsymbol{\theta}}(\boldsymbol{x}_t, t)$ denotes the decoded reconstruction, and $\boldsymbol{U}_k^\star$ is the basis matrix for the true class data subspace.

This formulation measures how well the learned feature emphasizes the signal components aligned with the true class subspace versus those aligned with irrelevant or confounding directions. Intuitively, the numerator captures the energy of the feature projected onto the correct class subspace, while the

denominator measures the residual energy after removing this component from the reconstructed signal. A higher SNR value at a given noise level $t$ indicates that the representation $\hat{h}_{\theta}$ encodes more discriminative structure with respect to the true class, implying better alignment with the downstream classification objective.

## 3.4 Main theoretical results

Based upon the aforementioned setup, we show the following results to explain the unimodal representation dynamics.

**Proposition 1.** *Suppose the data $\boldsymbol{x}_0$ is drawn from a noisy `MoLRG` data distribution with $K$-class and noise level $\delta$ introduced in Assumption 1. Then the optimal $\{\boldsymbol{U}\}$ minimizing the loss (2) is the ground truth basis defined in (3), and the optimal DAE $\hat{\boldsymbol{x}}_{\boldsymbol{\theta}}^{\star}(\boldsymbol{x}_t, t)$ admits the analytical form:*

$$\hat{\boldsymbol{x}}_{\boldsymbol{\theta}}^{\star}(\boldsymbol{x}_t, t) = \sum_{l=1}^{K} w_l^{\star}(\boldsymbol{x}_t, t) \left( \zeta_t \boldsymbol{U}_l^{\star} \boldsymbol{U}_l^{\star\top} + \xi_t \widetilde{\boldsymbol{U}}_l^{\star} \widetilde{\boldsymbol{U}}_l^{\star\top} \right) \boldsymbol{x}_t, \tag{7}$$

*where $w_l^{\star}(\boldsymbol{x}_t, t)$ are the coefficients in (5) when $\{\boldsymbol{U}\} = \{\boldsymbol{U}_l^{\star}\}_{l=1}^{K}$.*

**Link to the fine-to-coarse generation shift.** Since the $\boldsymbol{U}_l^{\star}$-related component captures low-dimensional class-relevant attributes and the $\widetilde{\boldsymbol{U}}_l^{\star}$-related component captures small-scale, class-irrelevant attributes, the optimal DAE exhibits a fine-to-coarse transition [46–48] in its output, where the class-irrelevant attributes are progressively removed as the noise level $\sigma_t$ increases. Specifically, $\zeta_t = \frac{1}{1+\sigma_t^2}$ quantifies the reduction rate of $\boldsymbol{U}_l^{\star}$ term while $\xi_t = \frac{\delta^2}{\delta^2+\sigma_t^2}$ quantifies the reduction rate of $\widetilde{\boldsymbol{U}}_l^{\star}$ term, as $\sigma_t$ grows, $\xi_t$ decays much more rapidly than $\zeta_t$, indicating that the output retains more class-related coarse information while discarding fine-grained, irrelevant details.

This phenomenon correspond to an important observation formalized in the next theorem: there exists a balance timestep during the diffusion process, at which class-irrelevant components are maximally suppressed while class-relevant component is preserved, yielding peak classification accuracy from the feature. Substituting the optimal DAE formulation into (6), we can approximate the SNR in the following theorem and analyze the unimodal dynamics via the approximation:

**Theorem 1.** *(Informal) Suppose that the data $\boldsymbol{x}_0$ follows the noisy `MoLRG` introduced in Assumption 1. Then the SNR of the optimal DAE $\hat{\boldsymbol{x}}_{\boldsymbol{\theta}}^{\star}$ can be approximated as follows:*

$$\mathrm{SNR}(\hat{\boldsymbol{x}}_{\boldsymbol{\theta}}^{\star}, t) \approx \frac{C_t}{(K-1)} \cdot \left( \frac{1 + \frac{\sigma_t^2}{\delta^2} h(\hat{w}_t^+, \delta)}{1 + \frac{\sigma_t^2}{\delta^2} h(\hat{w}_t^-, \delta)} \right)^2. \tag{8}$$

*Here, $C_t$ is a monotonically decaying constant that has minimal impact to the overall unimodal shape. The function $h(w, \delta) := (1 - \delta^2)w + \delta^2$ is monotonically increasing in $w$, where $h(\hat{w}_t^+, \delta)$ and $h(\hat{w}_t^-, \delta)$ denote positive and negative class confidence rates with*

$$\begin{cases} \hat{w}_t^+(\sigma_t, \delta) &= \mathbb{E}_k[\mathbb{E}_{\boldsymbol{x}_t}[w_k^{\star}(\boldsymbol{x}_t, t) \mid k]], \\ \hat{w}_t^-(\sigma_t, \delta) &= \mathbb{E}_k[\mathbb{E}_{\boldsymbol{x}_t}[w_l^{\star}(\boldsymbol{x}_t, t) \mid k \neq l]], \end{cases}$$

*which are the softmax coefficients assigned to the correct class $k$ and the other classes $l \neq k$.*

We defer the formal statement of Theorem 1 and its proof to Appendix E.2. In the following, we discuss the implications of our result.

**The unimodal curve of SNR across noise levels.** Intuitively, our theorem shows that a unimodal curve is mainly induced by the interplay between the "denoising rate" $\sigma_t^2/\delta^2$ and the positive class confidence rate $h(\hat{w}_t^+, \delta)$ as the noise level $\sigma_t$ increases. As observed in Figure 4, the "denoising rate" $(\sigma_t^2/\delta^2)$ increases monotonically with $\sigma_t$, while the class confidence rate $h(\hat{w}_t^+, \delta)$ monotonically declines. Initially, when $\sigma_t$ is small, the class confidence rate remains relatively stable due to its flat slope, and an increasing "denoising rate" improves the SNR. However, as indicated by (7), when $\sigma_t$ becomes too large, $h(\hat{w}_t^+, \delta)$ quickly decays and approaches $h(\hat{w}_t^-, \delta)$, leading to a drop in the SNR. It is worth noting that though we are mainly characterizing the unimodal dynamics of an approximation of SNR, in practice it closely mimic the trend of the actual SNR as in Figure 3b.

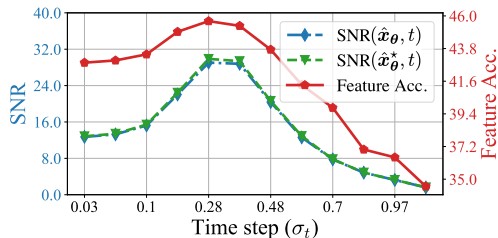
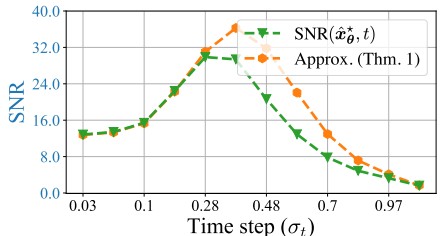

(a) SNR aligns with feature probing accuracy

(b) Tightness of the derived approximation

Figure 3: **Feature probing accuracy and associated** SNR **dynamics in** `MoLRG` **data.** In panel(a) we plot the probing accuracy and SNR with the feature obtained from a learned DAE $\hat{x}_\theta$, both of which exhibit a consistent unimodal pattern. The DAE is trained on a 3-class `MoLRG` dataset with data dimension $n = 50$, subspace dimension $d = 5$, and noise scale $\delta = 0.2$. Additionally, in panel(b) we include the optimal SNR calculated from the optimal DAE $\hat{x}_\theta^\star$ and the derived approximation in Theorem 1 as a reference.

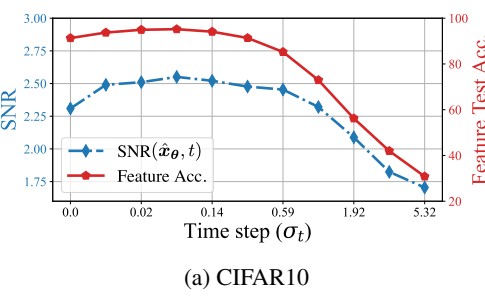
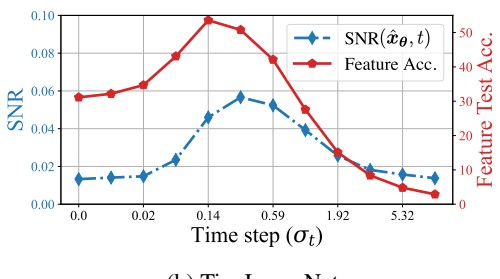

(a) CIFAR10

(b) TinyImageNet

Figure 5: **Dynamics of feature accuracy and associated** SNR **on CIFAR10 and TinyImageNet.** Feature accuracy is plotted alongside $\mathrm{SNR}(\hat{x}_\theta, t)$. Feature accuracy is evaluated on the test set, while the empirical SNR is computed from the training set. Both exhibit an aligning unimodal pattern. We use released EDM models [49] trained on the CIFAR10 [50] and ImageNet [51] datasets, evaluating them on CIFAR10 and TinyImageNet [52], respectively. To compute SNR , we apply PCA on the CIFAR10/TinyImageNet features to extract the basis $U_l$s. Further details can be found in Appendix D.

**Alignment of** SNR **with representation learning performance.** As shown in Figures 3 and 5, our theory derived from the noisy `MoLRG` distribution effectively captures real-world phenomena. Specifically, we conduct experiments on both synthetic (i.e., noisy `MoLRG`) and real-world datasets (i.e., CIFAR and ImageNet) to measure $\mathrm{SNR}(\hat{x}_\theta, t)$ as well as the feature probing accuracy. For feature probing, we use features extracted at different timesteps as inputs for linear probing. The results consistently show that $\mathrm{SNR}(\hat{x}_\theta, t)$ follows a unimodal pattern across all cases, mirroring the trend observed

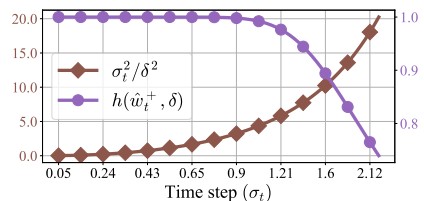

Figure 4: Illustration of the interplay between the denoising rate and the class confidence rate. The settings follow Figure 3.

in feature probing accuracy as the noise scale increases. This alignment provides a formal justification for previous empirical findings [19, 20, 23], which have reported a unimodal trajectory in the representation dynamics of diffusion models with increasing noise levels. Detailed experimental setups are provided in Appendix D.

## 4   Unimodal representation dynamics predicts model generalization

In the previous sections, we theoretically showed that when a diffusion model successfully captures the low-dimensional distribution of the data, the unimodal representation dynamics emerge. In

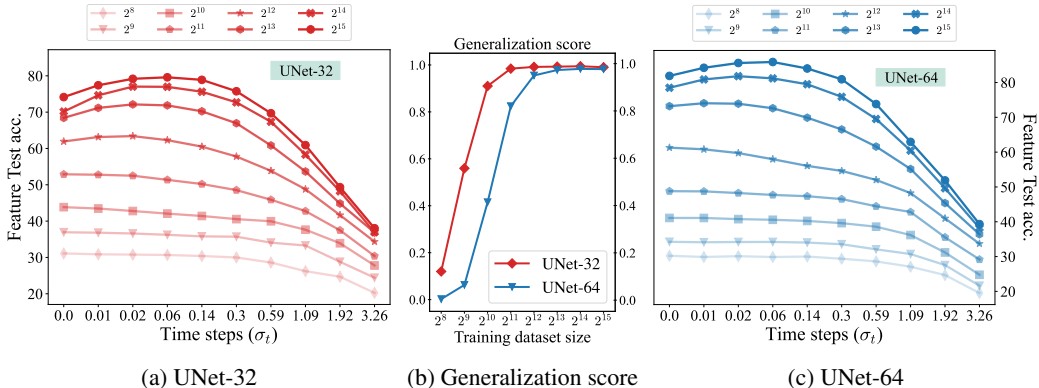

(a) UNet-32  (b) Generalization score  (c) UNet-64

Figure 6: **Representation dynamics across model and data sizes.** We train DDPM-based UNet-32 and Unet-64 diffusion models on the CIFAR10 dataset using different training dataset sizes, ranging from $2^8$ to $2^{15}$. The unimodal representation dynamics across noise levels consistently emerges in the generalization regime (sufficient data size) and gradually disappears in smaller data settings.

this section, we investigate the opposite direction: can the presence of the unimodal representation dynamics serve as a reliable prediction of good generalization of diffusion models?

Answering this question sheds light on the distribution learning capabilities of diffusion models and is closely related to recent studies on their generalizability, which reveal that diffusion models operate in two distinct regimes—generalization and memorization—depending on factors such as dataset size, model capacity, and training duration [32, 53–56]. In the generalization regime, the model captures the underlying data distribution and generates diverse, novel samples. In contrast, in the memorization regime, it overfits to the training data and loses the ability to generate novel samples. Further discussion on the generalization of diffusion models are provided in Appendix A.

In this section, using classification tasks as a case study, we empirically demonstrate that the presence of unimodal representation dynamics reliably indicates generalization, while its gradual shift to a monotonically decreasing trend indicates memorization. Specifically, we study the effects of data size and model capacity in Section 4.1, and the effects of learning dynamics in Section 4.2.

## 4.1 Effects of dataset size and model capacity on representation dynamics

Recent studies [31, 53] have shown that diffusion models exhibit a phase transition from memorization to generalization as the number of training samples increases. Specifically, when the network size significantly exceeds the number of training samples, diffusion models tend to memorize rather than capture the underlying low-dimensional data distribution [32, 53]. For fixed model capacity, generalization typically emerges when the number of training data is larger than a certain threshold; see Figure 6 (b). Here, we demonstrate that the representation dynamics undergo a similar transition with data scaling. Specifically, we train UNet-32 and UNet-64 [43] models using varying training dataset sizes to examine how their representation dynamics change across regimes. The generalization score metric introduced in [53, 57] is used as a reference for quantifying model generalization.

As shown in Figure 6, we observe that reducing the training dataset size leads to a decrease in the generalization score. Correspondingly, the unimodal representation dynamic becomes less obvious, eventually transitioning into a monotonically decreasing curve (i.e., *monotonic representation dynamics*). These observations highlight a strong connection between representation and distribution learning in diffusion models—specifically, the emergence of unimodal representation dynamics aligns with the ability of the model to capture the underlying data distribution for achieving good generalization.

## 4.2 Effects of learning dynamics on representation dynamics

Second, we investigate how learning dynamics influence representation dynamics and also generalization performance, particularly in the limited data regime (e.g., training size $N = 2^{12}$ as shown in Figure 6 (b)). In this regime, recent studies [54–56] have shown that *early stopping* can improve

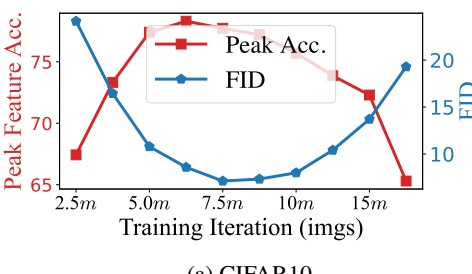 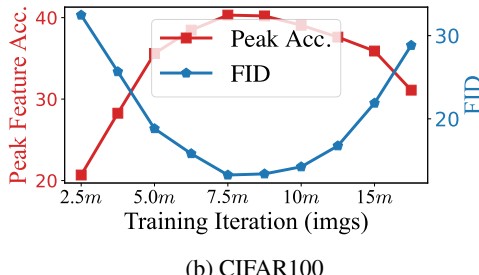

| (a) CIFAR10 | (b) CIFAR100 |
|---|---|

Figure 7: **Negative correlation between peak classification accuracy and FID.** We train UNet-128 diffusion models on $N = 2^{12}$ training samples from CIFAR10 and CIFAR100. As training progresses, the peak representation accuracy across noise levels shows a consistent negative correlation with FID.

generalization performance. Specifically, these works observe an early learning phenomenon, where generalization improves during the initial phase of training but deteriorates as the model begins to memorize. As illustrated in Figure 7, this effect is reflected in the evolution of the FID score [58], which initially decreases (indicating better generative quality) and then rises as memorization starts. Notably, we find that this trend negatively correlates with the linear probing accuracy of learned representations. This observation implies that representation quality could potentially serve as an early-stopping criterion to prevent memorization in diffusion models trained on limited data without relying on external models.

Moreover, we show that this early learning behavior can be captured by the transition between unimodal and monotonic curves of representation dynamics during training. Experimentally, we demonstrate this by training EDM-based diffusion models on subsets of CIFAR[50] using $N = 2^{12}$ training samples. Specifically, from Figure 8, we observe that the evolution of representation dynamics during training can be divided into two distinct phases:

- *Early phase of unimodal representation dynamics.* In the early stage of training, the model learns the underlying low-dimensional data distribution and is able to generalize. As predicted by our theoretical analysis in Section 3, representation follows a unimodal dynamic across noise scales. This unimodal dynamic is clearly observed before training iteration $\text{Iter} \leq 7.5M$ in Figure 8 (a). The generalization behavior is further supported by the new outputs of the model as shown in Figure 8 (b), which more closely resemble those from a reference generalized model than the nearest neighbors in the training set. Moreover, the peak representation quality improves steadily during this phase as the model better captures the data distribution.

- *Late phase of monotonic representation dynamics.* However, as training progresses toward convergence, the model begins to memorize the training samples, resulting in a reduced ability to capture the underlying data distribution. This transition is obvious in the outputs of the model, which increasingly replicate training examples (see Figure 8 (b) at $\text{Iter} = 15M, 100M$). During this phase, the unimodal representation dynamics give way to a monotonic representation dynamics, where representation quality consistently degrades as the noise level increases. Furthermore, as shown in Figure 7, the learned features become less informative, and the peak probing accuracy begins to decline.

## 5 Conclusion

In this work, we developed a mathematical framework for analyzing the representation dynamics of diffusion models. By introducing the concept of $\text{SNR}$ under a mixture of low-rank Gaussians, we showed that the widely observed unimodal representation dynamic across noise scales emerges naturally when diffusion models capture the underlying data distribution. This behavior arises from a trade-off between denoising strength and class confidence across noise levels. Beyond theoretical insights, our empirical results demonstrate that the emergence of unimodal representation dynamics is closely linked to the model's distribution learning and generalization ability. Specifically, this unimodal pattern consistently appears when the model generalizes and gradually fades as the model starts to memorize. Our findings take a step toward bridging the gap between generative modeling and representation learning in diffusion models.

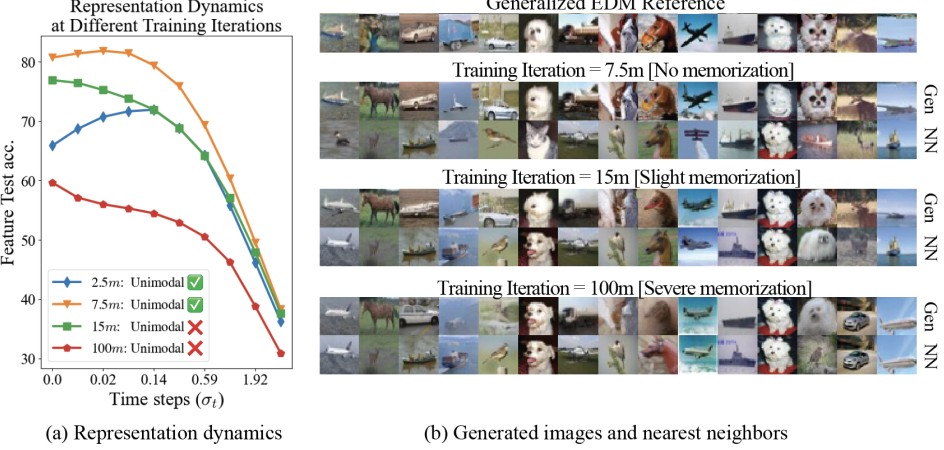

Figure 8: **Representation learning and generative performance across training iterations.** We train a UNet-128 diffusion model on $N = 2^{12}$ training samples in CIFAR10, monitoring both representation learning and generative performance as training progresses. A clear phase transition is observed: early in training, the representations exhibit a unimodal pattern, and generated samples resemble those of a generalizing EDM model, with no signs of memorization. As training continues, the unimodal pattern gradually transitions to a monotonically decreasing trend, aligning with the model's shift toward memorizing the training data. "NN" denotes nearest neighbor in the training dataset.

While our analysis captures key aspects of representation dynamics, it relies on simplified data assumptions and model formulations that facilitate tractable derivations. Extending this framework to more realistic data distributions and complex model architectures remains an important direction for future work. Moreover, establishing a rigorous theoretical connection between the representation geometry of diffusion models and their phase transition between generalization and memorization remains an important and promising direction for future work.

## Acknowledgment

We acknowledge funding support from NSF CAREER CCF-2143904, NSF CCF-2212066, NSF CCF- 2212326, NSF IIS 2312842, NSF IIS 2402950, NSF IIS 2312840, NSF IIS 2402952, ONR N00014-22-1-2529, ONR N000142512339, and MICDE Catalyst Grant. We also thank all the anonymous reviewers for their valuable suggestions and fruitful discussions.

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

# A    Related works

**Denoising auto-encoders.**    Denoising autoencoders (DAEs) are trained to reconstruct corrupted images to extract semantically meaningful information, which can be applied to various vision [59, 60] and language downstream tasks [61]. Related to our analysis of the weight-sharing mechanism, several studies have shown that training with a noise scheduler can enhance downstream performance [62–64]. On the theoretical side, prior works have studied the learning dynamics [65, 66] and optimization landscape [67] through the simplified linear DAE models.

**Diffusion-based representation learning.**    Diffusion-based representation learning [68] has demonstrated significant success in various downstream tasks, including image classification [20, 21, 69], segmentation [19], correspondence [23], and image editing [70]. Using diffusion models for data augmentation has also been shown to improve robustness against covariate shift [71]. To further enhance the utility of diffusion features, knowledge distillation [72–75] methods have been proposed, aiming to bypass the computationally expensive grid search for the optimal $t$ in feature extraction and improving downstream performance. Beyond directly using intermediate features from pre-trained diffusion models, research efforts has also explored novel loss functions [76, 77] and network modifications [78, 79] to develop more unified generative and representation learning capabilities within diffusion models. The work [80] investigates whether diffusion models are capable of learning latent inter-feature dependencies underlying image data. Unlike the aforementioned efforts, our work focuses more on understanding the representation dynamic of diffusion models and its relationship with model generalization.

**Generalization of Diffusion Models.**    In this paper, we examine the relationship between a unimodal curve in representation-probing accuracy and the generalization of diffusion models, a topic of active interest [31, 48, 53]. There has also been notable work on the mechanisms that enable diffusion models to learn the underlying score function from discrete empirical samples [81, 82], thereby generating novel in-distribution samples (i.e., *generalizing*). We propose a representational perspective on the generalization of diffusion models, together with corresponding measures (unimodality in representation dynamics) for evaluating generalization.

The unimodality of representation-probing accuracy across time steps is also related to symmetry breaking [83] and early-stopping generalization during sampling [84, 85] when the model learns empirical score functions. This line of work further connects empirical DMs and dense associative memory (AM) networks through their objectives and sampling behavior [86–88], where generalization is interpreted as novel attraction sinks in AM, which is also related to Gaussian and spherical patterns in the data. However, in this paper our focus is on learning a parameterized model with data from a low-dimensional distribution, an assumption also used in [89, 90] (which links generalization to adapting to low-dimensional data manifolds and learning local geometric components), rather than on sampling.

# B    Auxiliary results

## B.1    Extended results from Section 4.2

In Section 4.2, we showed that when training on limited data, the representation dynamics undergo a clear phase transition: from a unimodal pattern to a monotonically decreasing trend, where the diffusion model also exhibits a transition from generalization to increasingly memorize the training data. Here, we provide additional empirical evidence supporting this insight.

We train UNet-based diffusion models using the DDPM++ architecture with the EDM configuration [49] on the Oxford-IIIT Pet [91] and TinyImageNet [92] datasets, using training subsets of 3680 and 2048 images, respectively. Throughout training, we monitor both the evolution of representation dynamics and generative outputs. As shown in Figure 9 and Figure 10, consistent with our findings

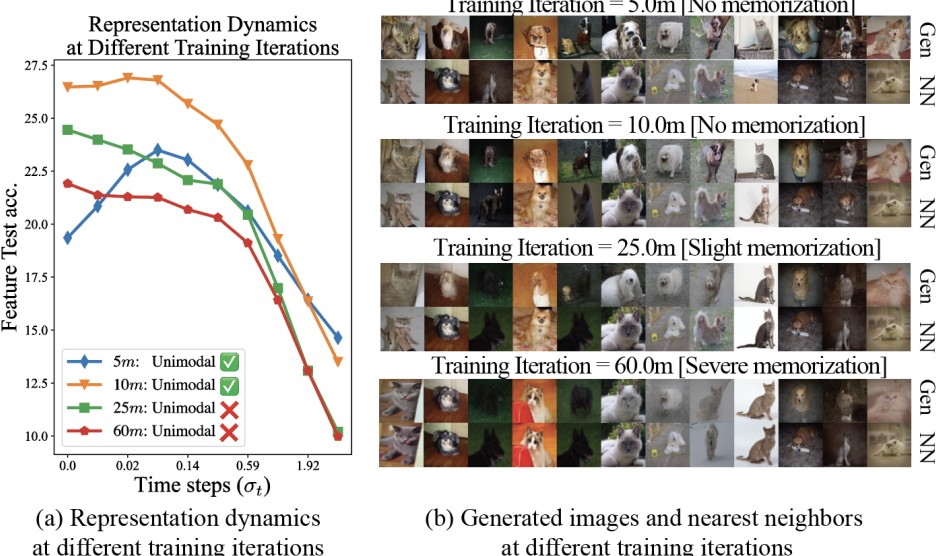

(a) Representation dynamics
at different training iterations

(b) Generated images and nearest neighbors
at different training iterations

Figure 9: **Representation learning and generative performance across training iterations for Oxford-IIIT Pet dataset [91].** We train a UNet-128 diffusion model on 3680 training samples in Oxford-IIIT Pet dataset, monitoring both representation learning and generative performance as training progresses. A clear phase transition is observed: early in training, the representations exhibit a unimodal pattern, and generated samples resemble those of a generalizing EDM model, with no signs of memorization. As training continues, the unimodal pattern gradually transitions to a monotonically decreasing trend, aligning with the model's shift toward memorizing the training data. "NN" denotes nearest neighbor in the training dataset.

in Section 4.2, we observe that in the early stages of training, the model exhibits a clear unimodal pattern and generalizes well. As training continues, this unimodal structure gradually shifts into a monotonically decaying trend, and the models start to increasingly replicate training examples.

## B.2 Disentangling the role of input noise in representation dynamics

One might argue that the declining portion of the unimodal curve is simply due to the increasing noise level $\sigma_t$, which makes the input $x_t$ progressively noisier, leading to a natural drop in classification accuracy. However, we show that this noise-induced degradation alone does not account for the observed representation dynamics.

Our theoretical analysis in Section 3 attributes the unimodal pattern to a fundamental trade-off between denoising rate and class confidence rate across noise levels. To validate this explanation and disentangle the effect of additive Gaussian noise, we conduct experiments where feature extraction is performed directly on clean inputs $x_0$ rather than noisy inputs $x_t$. We show that the unimodal behavior remains clearly observable even in the absence of injected noise, indicating that the dynamics are not solely a consequence of input corruption. In fact, one can verify that $\mathrm{SNR}(\hat{x}_{\theta}^{\star}(x_0), t)$ also exhibits a unimodal trend—this can be shown through an analysis analogous to that of Theorem 1 for $\mathrm{SNR}(\hat{x}_{\theta}^{\star}(x_t), t)$.

To support this claim, we train EDM (with the VP configuration) [49] on CIFAR datasets and classical DDPM [2] on CIFAR10 and Flowers-102 [93], and evaluate the representation quality using both noisy inputs $x_t$ and clean inputs $x_0$, as shown in Figure 11. For Figure 11(a), we select the layer that yields the best accuracy, while for Figure 11(b), we directly use the bottleneck layer to demonstrate that the observed unimodal behavior is not sensitive to layer choice. Across all settings, the unimodal representation dynamics remain clearly visible, reinforcing that additive Gaussian noise is not the sole factor responsible for this phenomenon.

We also visualize posterior estimation $\hat{x}_{\theta}$ across noise scales using both noisy and clean inputs. Since diffusion models are trained to approximate the posterior mean at different noise levels,

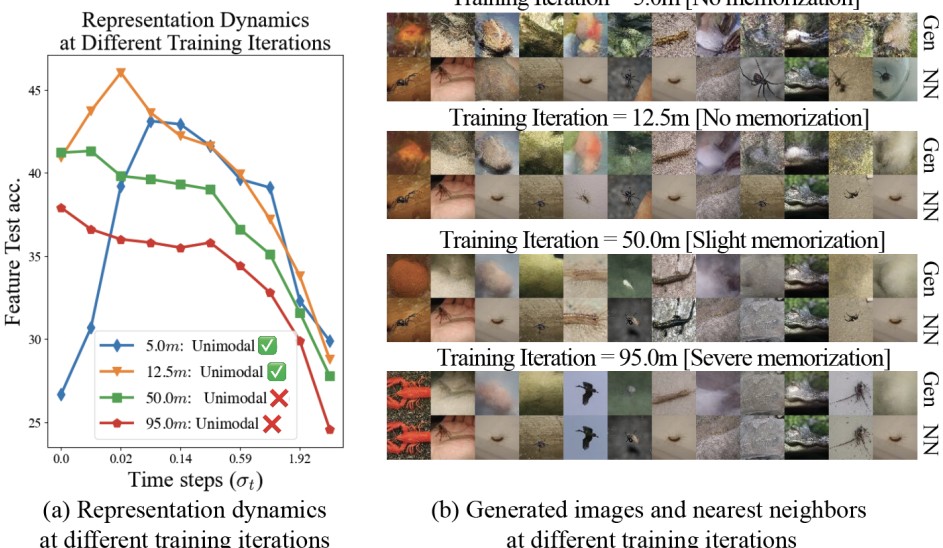

(a) Representation dynamics
at different training iterations

(b) Generated images and nearest neighbors
at different training iterations

Figure 10: **Representation learning and generative performance across training iterations for TinyImageNet [92].** We train a UNet-128 diffusion model on $N = 2^{11}$ training samples in TinyImageNet, monitoring both representation learning and generative performance as training progresses. A clear phase transition is observed: early in training, the representations exhibit a unimodal pattern, and generated samples resemble those of a generalizing EDM model, with no signs of memorization. As training continues, the unimodal pattern gradually transitions to a monotonically decreasing trend, aligning with the model's shift toward memorizing the training data. "NN" denotes nearest neighbor in the training dataset.

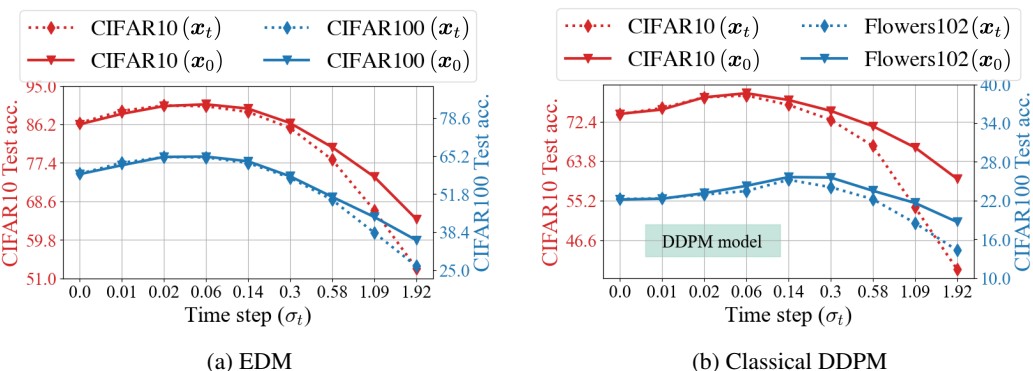

(a) EDM

(b) Classical DDPM

Figure 11: **Unimodal representation dynamics persist when using clean inputs $x_0$.** We train EDM and DDPM models on CIFAR and Flowers-102 datasets and evaluate feature quality using both noisy inputs $x_t$ and clean inputs $x_0$. Across all settings, the unimodal trend is consistently observed.

their representation features emerge as intermediate products of this denoising process. As such, improvements or degradations in representation quality should be mirrored in the posterior estimates.

We visualize the posterior estimation results for clean inputs $(\hat{x}_{\theta}(x_0, t))$ and noisy inputs $(\hat{x}_{\theta}(x_t, t))$ across varying noise scales $\sigma_t$ in Figure 12. In both cases, the posterior outputs undergo a clear fine-to-coarse transition as $\sigma_t$ increases. This supports our theoretical claim that as noise grows, class-irrelevant attributes are gradually removed. The peak in representation quality occurs at an intermediate point where class-essential structure is preserved while irrelevant details are suppressed. When $\sigma_t$ becomes too large, class confidence rate drops significantly, resulting in poor representations. The additive noise $\sigma_t \epsilon$ merely accelerates this degradation but is not the root cause.

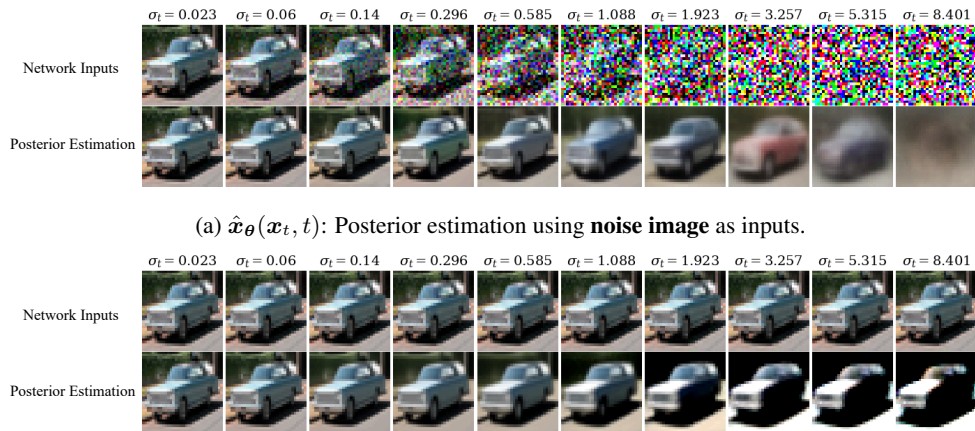

(a) $\hat{\boldsymbol{x}}_{\boldsymbol{\theta}}(\boldsymbol{x}_t, t)$: Posterior estimation using **noise image** as inputs.

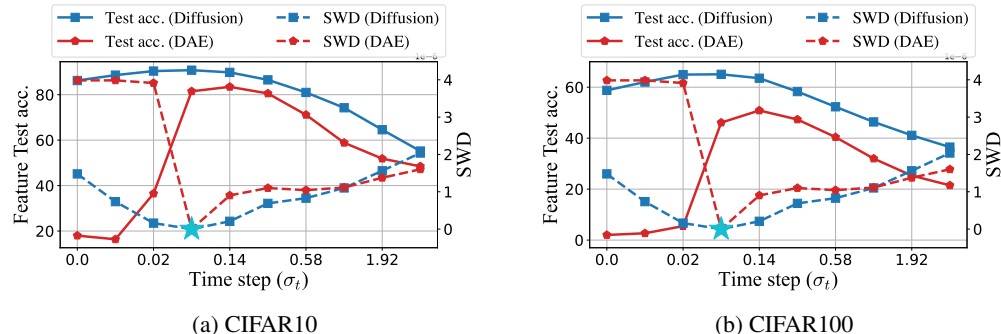

(b) $\hat{\boldsymbol{x}}_{\boldsymbol{\theta}}(\boldsymbol{x}_0, t)$: Posterior estimation using **clean image** as inputs.

Figure 12: **Fine-to-coarse shift in posterior estimation.** We use a pre-trained DDPM diffusion model on CIFAR10 to visualize posterior estimation for clean inputs and noisy inputs across varying noise scales $\sigma_t$. We can observe seemingly fine-to-coarse shifts in both figures.

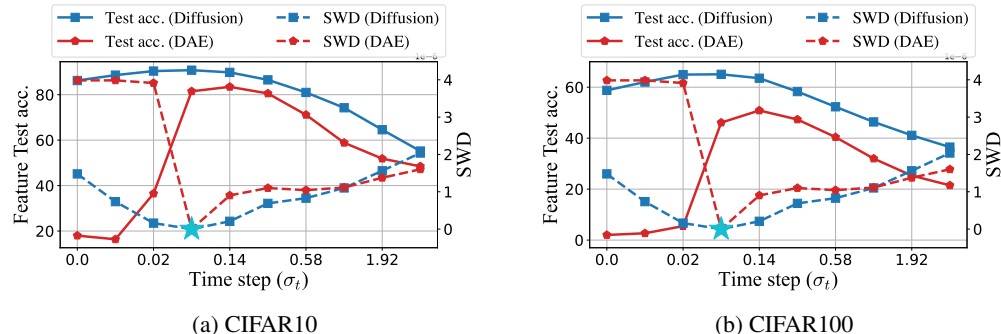

(a) CIFAR10                    (b) CIFAR100

Figure 13: **Diffusion models exhibit higher and smoother feature accuracy and similarity compared to individual DAEs.** We train DDPM-based diffusion models and individual DAEs on the CIFAR datasets and evaluate their representation learning performance. Feature accuracy, and feature differences from the optimal features (indicated by ⋆) are plotted against increasing noise levels. The results reveal an inverse correlation between feature accuracy and feature differences, with diffusion models achieving both higher/smoother accuracy and smaller/smoother feature differences compared to DAEs.

## B.3  Weight sharing in diffusion models facilitates representation learning

While our theoretical analysis captures the emergence of unimodal representation dynamics under an idealized network parameterization, an important future direction is to extend this framework to deeper and more complex architectures. Real-world diffusion models often involve highly complex feature transformations, and understanding how these interact with noise scales to influence representation quality remains an open and valuable avenue for exploration.

In this section, we present some interesting preliminary results we found that may potentially explain why diffusion models outperform classical denoising autoencoders (DAEs) in representation learning: although both share the same denoising objective (2), diffusion models demonstrate superior feature learning capabilities largely due to their inherent weight-sharing mechanism. Specifically, by minimizing the loss across all noise levels, diffusion models enable parameter sharing and interaction among denoising subcomponents, effectively creating an implicit "ensemble" effect. This interaction enhances feature consistency and robustness across noise scales, leading to significantly improved representation quality compared to DAEs [22], as illustrated in Figure 13.

| Method | *MiniImageNet** Test Acc. % | | | | |
|--------|-------|------|------|------|------|
| **Label Noise** | Clean | 20% | 40% | 60% | 80% |
| MAE | 73.7 | 70.3 | 67.4 | 62.8 | 51.5 |
| EDM | 67.2 | 62.9 | 59.2 | 53.2 | 40.1 |
| **EDM (Ensemble)** | 72.0 | 67.8 | 64.7 | 60.0 | 48.2 |
| DiT | 77.6 | 72.4 | 68.4 | 62.0 | 47.3 |
| **DiT (Ensemble)** | **78.4** | **75.1** | **71.9** | **66.7** | **56.3** |

Table 1: **Comparison of test performance across different methods under varying label noise levels.** All compared models are publicly available and pre-trained on ImageNet-1K [51], evaluated using MiniImageNet classes. Bold font highlights the best result in each scenario.

To test this, we trained 10 DAEs, each specialized for a single noise level, alongside a DDPM-based diffusion model on CIFAR10 and CIFAR100. We compared feature quality using linear probing accuracy and feature similarity relative to the optimal features at $\sigma_t = 0.06$ (where accuracy peaks) via sliced Wasserstein distance (SWD) [94].

The results in Figure 13 confirm the advantage of diffusion models over DAEs. Diffusion models consistently outperform DAEs, particularly in low-noise regimes where DAEs collapse into trivial identity mappings. In contrast, diffusion models leverage weight-sharing to preserve high-quality features, ensuring smoother transitions and higher accuracy as noise increases. This advantage is further supported by the SWD curve, which reveals an inverse correlation between feature accuracy and feature differences. Notably, diffusion model features remain significantly closer to their optimal state across all noise levels, demonstrating superior representational capacity.

Our finding also aligns with prior results that sequentially training DAEs across multiple noise levels improves representation quality [62–64]. Our ablation study further confirms that multi-scale training is essential for improving DAE performance on classification tasks in low-noise settings (details in Appendix C, Table 3).

Beyond the implicit feature ensembling effect, we further introduce a straightforward method that explicitly ensembles features from multiple noise levels to enhance downstream task performance. Our experiments demonstrate that this approach significantly improves robustness against label noise in classification tasks, both in pre-training and transfer learning settings. For detailed methods and results, we refer interested readers to Appendix B.4.

### B.4 Feature ensembling across timesteps improves representation robustness

Our theoretical insights in Section 3 imply that features extracted at different timesteps capture varying levels of granularity. Given the high linear separability of intermediate features, we propose a simple ensembling approach across multiple timesteps to construct a more holistic representation of the input. Specifically, in addition to the optimal timestep, we extract feature representations at four additional timesteps—two from the coarse (larger $\sigma_t$) and two from the fine-grained (smaller $\sigma_t$) end of the spectrum. We then train linear probing classifiers for each set and, during inference, apply a soft-voting ensemble by averaging the predicted logits before making a final decision.(experiment details in Appendix D)

We evaluate this ensemble method against results obtained from the best individual timestep, as well as a self-supervised method MAE [95], on both the pre-training dataset and a transfer learning setup. The results, reported in Table 1 and Table 2, demonstrate that ensembling significantly enhances performance for both EDM [49] and DiT [37], consistently outperforming their vanilla diffusion model counterparts and often surpassing MAE. More importantly, ensembling substantially improves the robustness of diffusion models for classification under label noise.

## C   Additional Experiments

**Validation of $\hat{x}^\star_{approx}$ approximation in Appendix E.2.**   In Theorem 2, we approximate the optimal posterior estimation function $\hat{x}^\star_\theta$ using $\hat{x}^\star_{approx}$ by taking the expectation inside the softmax with respect to $x_t$. To validate this approximation, we compare the SNR calculated from $\hat{x}^\star_\theta$ and from $\hat{x}^\star_{approx}$ using the definition in Proposition 1 and (10) in Appendix E.2, respectively. We use a fixed

| Method | Transfer Test Acc. % | | | | | | | | | | | | | | |
|---|---|---|---|---|---|---|---|---|---|---|---|---|---|---|---|
| | **CIFAR100** | | | | | **DTD** | | | | | **Flowers102** | | | | |
| **Label Noise** | Clean | 20% | 40% | 60% | 80% | Clean | 20% | 40% | 60% | 80% | Clean | 20% | 40% | 60% | 80% |
| MAE | 63.0 | 58.8 | 54.7 | 50.1 | 38.4 | 61.4 | 54.3 | 49.9 | 40.5 | 24.1 | 68.9 | 55.2 | 40.3 | 27.6 | 9.6 |
| EDM | 62.7 | 58.5 | 53.8 | 48.0 | 35.6 | 54.0 | 49.1 | 45.1 | 36.4 | 21.2 | 62.8 | 48.2 | 37.2 | 24.1 | 9.7 |
| **EDM (Ensemble)** | **67.5** | **64.2** | **60.4** | **55.4** | **43.9** | 55.7 | 49.5 | 45.2 | 37.1 | 22.0 | 67.8 | 53.9 | 41.5 | 25.0 | 10.4 |
| DiT | 64.2 | 58.7 | 53.5 | 46.4 | 32.6 | 65.2 | 59.7 | 53.0 | 43.8 | 27.0 | 78.9 | 65.2 | 52.4 | 34.7 | 13.3 |
| **DiT (Ensemble)** | 66.4 | 61.8 | 57.6 | 51.3 | 39.2 | **65.3** | **60.6** | **56.1** | **46.3** | **30.6** | **79.7** | **67.0** | **54.6** | **36.6** | **14.7** |

Table 2: **Comparison of transfer learning performance across different methods under varying label noise levels.** All compared models are publicly available and pre-trained on ImageNet-1K [51], evaluated on different downstream datasets. Bold font highlights the best result in each scenario.

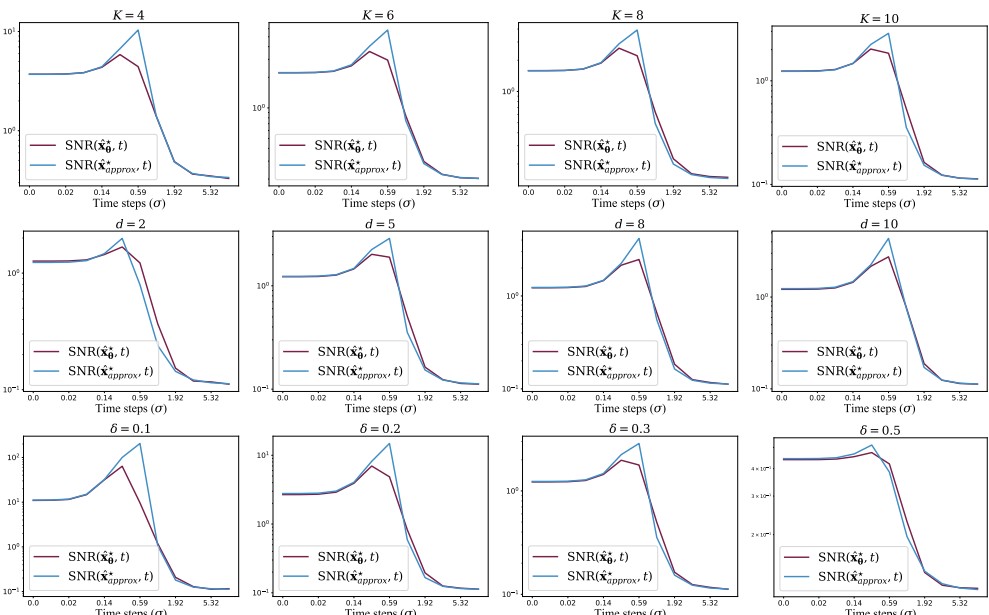

Figure 14: **Comparison between** SNR **calculated using the optimal model** $\hat{x}_{\theta}^{\star}$ **and the** SNR **calculated with our approximation in Theorem 1.** We generate MoLRG data and calculate SNR using both the corresponding optimal posterior function $\hat{x}_{\theta}^{\star}$ and our approximation $\hat{x}_{approx}^{\star}$ from Theorem 1. Default parameters are set as $n = 100$, $d = 5$, $K = 10$, and $\delta = 0.3$. In each row, we vary one parameter while keeping the others fixed, comparing the actual and approximated SNR.

dataset size of 2400 and set the default parameters to $n = 100$, $d = 5$, $K = 10$, and $\delta = 0.3$ to generate MoLRG data. We then vary one parameter at a time while keeping the others constant, and present the computed SNR in Figure 14. As shown, the approximated SNR score consistently aligns with the actual score.

**Visualization of the** MoLRG **posterior estimation and** SNR **across noise scales.** In Figure 3, we show that both the classification accuracy and SNR exhibit a unimodal trend for the MoLRG data. To further illustrate this behavior, we provide a visualization of the posterior estimation and SNR at different noise scales in Figure 15. In the plot, each class is represented by a colored straight line, while deviations from these lines correspond to the $\delta$-related noise term. Initially, increasing the noise scale effectively cancels out the $\delta$-related data noise, resulting in a cleaner posterior estimation and improved probing accuracy. However, as the noise continues to increase, the class confidence rate drops, leading to an overlap between classes, which ultimately degrades the feature quality and probing performance.

**Mitigating the performance gap between DAE and diffusion models.** Throughout the empirical results presented in this paper, we consistently observe a performance gap between individual DAEs and diffusion models, especially in low-noise regions. Here, we use a DAE trained on the CIFAR10 dataset with a single noise level $\sigma = 0.002$, using the NCSN++ architecture [49]. In

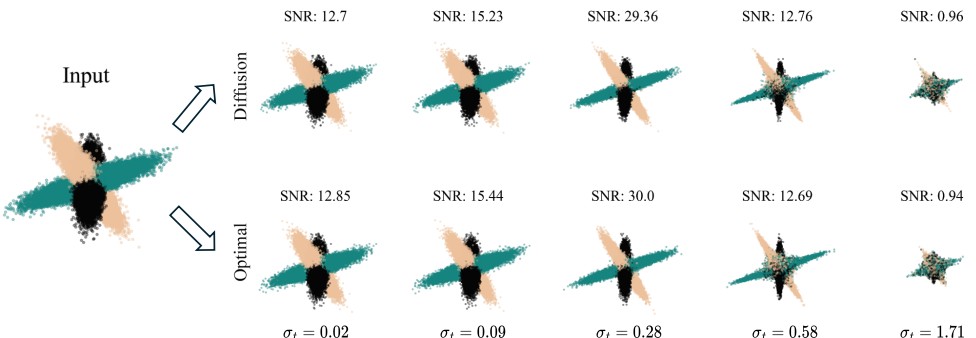

Figure 15: **Visualization of posterior estimation, higher** SNR **correspondings to higher classification accuracy.** The **same** `MoLRG` data is fed into the models; each row represents a different denoising model, and each column corresponds to a different time step with noise scale ($\sigma_t$).

Table 3: **Improve DAE representation performance at low noise region.** A vanilla DAE trained on the CIFAR10 dataset with a single noise level of $\sigma = 0.002$ serves as the baseline. We evaluate the performance improvement of dropout regularization, EDM-based preconditioning, and multi-level noise training ($\sigma = \{0.002, 0.012, 0.102\}$). Each technique is applied independently to assess its contribution to performance enhancement.

| Modifications | Test acc. |
|---|---|
| Vanilla DAE | 32.3 |
| +Dropout (0.5) | 35.3 |
| +Dropout (0.9) | 36.4 |
| +Dropout (0.95) | 38.1 |
| +EDM preconditioning | 49.2 |
| +Multi-level noise training | **58.6** |

the default setting, the DAE achieves a test accuracy of 32.3. We then explore three methods to improve the test performance: (a) adding dropout, as noise regularization and dropout have been effective in preventing autoencoders from learning identity functions [66]; (b) adopting EDM-based preconditioning during training, including input/output scaling, loss weighting, etc.; and (c) multi-level noise training, in which the DAE is trained simultaneously on three noise levels [0.002, 0.012, 0.102]. Each modification is applied independently, and the results are reported in Table 3. As shown, dropout helps improve performance, but even with a dropout rate of 0.95, the improvement is minor. EDM-based preconditioning achieves moderate improvement, while multi-level noise training yields the most promising results, demonstrating the benefit of incorporating the diffusion process in DAE training.

**Empirical robustness to assumption relaxation**   The data assumptions in Assumption 1 were made to simplify the analysis and derive closed-form results. Empirically, we find that the unimodal SNR trend remains robust when these assumptions are relaxed. To demonstrate this, we train our parameterized DAE under binary `MoLRG` data while systematically violating each assumption. Unless otherwise stated, each experiment uses the DAE network as introduced in Section 3.2 and a dataset contains 30,000 samples generated with $n = 50$, $d = 5$, and $\delta = 0.2$.

- *Overlapping class subspaces.* As shown in Table 4, we control the principal angle ($\theta$) between class subspaces and observe that overlap tends to reduce SNR across timesteps while the peak remains stable. We conjecture that overlap can be viewed as introducing additional intrinsic noise beyond the $\delta$-related term, thereby affecting the SNR value.

- *Varying subspace ranks.* As shown in Table 5, we set the class subspace dimensions to $d_0 = 10$ and $d_1 = 2$. Intuitively, the higher-rank class retains more signal and is less sensitive to noise, yielding higher and later-peaking SNR, while the low-rank class decays earlier.

- *Non-uniform mixing weights.* As shown in Table 6, we set $\pi_0 = 0.8$ and $\pi_1 = 0.2$ and observe consistently higher SNR for the majority class. We conjecture that this may stem from both the

score function of the distribution and DAE training being biased toward denoising more frequent samples.

| Training Setting | SNR @ $\sigma_t$ | | | | | | | |
|---|---|---|---|---|---|---|---|---|
| | 0.008 | 0.023 | 0.060 | 0.140 | 0.296 | 0.585 | 1.088 | 1.923 |
| $\theta = 90°$ (Non-overlapping) | 24.88 | 25.15 | 26.95 | 35.84 | 58.27 | 20.84 | 4.05 | 1.57 |
| $\theta = 30°$ (Overlapping) | 16.70 | 16.92 | 18.39 | 25.97 | 38.54 | 22.88 | 14.56 | 12.55 |

Table 4: **Effect of overlapping class subspaces on SNR across noise levels.**

| Training Setting | SNR @ $\sigma_t$ | | | | | | | | |
|---|---|---|---|---|---|---|---|---|---|
| | 0.002 | 0.008 | 0.023 | 0.060 | 0.140 | 0.296 | 0.585 | 1.088 | 1.923 |
| Class 0 ($d_0 = 10$) | 68.94 | 69.00 | 69.83 | 74.87 | 102.24 | 195.44 | 215.31 | 34.17 | 8.44 |
| Class 1 ($d_1 = 2$) | 7.75 | 7.75 | 7.80 | 8.08 | 8.79 | 6.50 | 1.80 | 0.46 | 0.27 |
| Overall SNR | 24.12 | 24.13 | 24.28 | 25.21 | 28.19 | 23.34 | 8.18 | 2.84 | 1.45 |

Table 5: **Effect of subspace rank variation on SNR across noise levels.**

| Training Setting | SNR @ $\sigma_t$ | | | | | | | | |
|---|---|---|---|---|---|---|---|---|---|
| | 0.002 | 0.008 | 0.023 | 0.060 | 0.140 | 0.296 | 0.585 | 1.088 | 1.923 |
| Class 0 ($\pi_0 = 0.8$) | 52.35 | 52.43 | 52.98 | 56.74 | 75.61 | 134.76 | 72.38 | 13.56 | 4.50 |
| Class 1 ($\pi_1 = 0.2$) | 12.15 | 12.16 | 12.31 | 13.10 | 17.49 | 23.91 | 6.05 | 1.17 | 0.52 |
| Overall SNR | 38.69 | 38.74 | 39.17 | 41.85 | 55.77 | 90.54 | 35.28 | 7.40 | 2.81 |

Table 6: **Effect of non-uniform class mixing on SNR across noise levels.**

**Well-separated clusters.** In our main analysis, we considered zero-mean Gaussian clusters. In this section, we show that when the clusters are well separated, the unimodal trend disappears: both classification accuracy and SNR decrease monotonically, peaking at $t = 0$. In this regime, the dataset remains linearly separable throughout the denoising process, and both metrics are primarily determined by the class means. Adding noise (increasing $t$) simply blurs the class boundaries, thereby reducing separability. To verify this, we conduct an experiment on well-separated data and report the results in Table 7, which clearly exhibit this monotonic behavior. These observations highlight the importance of overlapping clusters in giving rise to the unimodal representation dynamics.

**Unimodal dynamics beyond pixel space.** The unimodal dynamics are not restricted to pixel-space denoisers. In the table below, we analyze feature accuracy and SNR dynamics using the DiT-XL/2 [37] model on `miniImageNet`, and observe in Table 8 that the unimodal trend persists in this latent diffusion model setting as well.

**Projection-based classification analysis.** In the main paper, our linear probing experiments always use logistic regression classifiers with inner-product-based logits. Here, we re-ran the experiments in Figure 5 using a projection-based classifier: for each class, we compute its principal subspace $[\boldsymbol{V}_1, \ldots, \boldsymbol{V}_K]$ from the training features, and classify each test sample by identifying the class whose subspace captures the most projection energy, computed as $\arg\max_k \|\boldsymbol{h}(\boldsymbol{x}_i)\boldsymbol{V}_k\|^2$.

The results are shown in Table 9. While the projection-based classifier yields lower overall accuracy, it reveals a more pronounced unimodal trend that aligns closely with the SNR dynamics.

# D Experimental Details

In this section, we provide technical details for all the experiments in the main body of the paper.

| $\sigma_t$ | 0.030 | 0.053 | 0.098 | 0.189 | 0.282 | 0.379 | 0.480 | 0.588 | 0.704 | 0.830 | 0.989 | 1.492 |
|---|---|---|---|---|---|---|---|---|---|---|---|---|
| **Acc** | 100.0 | 100.0 | 100.0 | 99.8 | 97.7 | 92.4 | 85.7 | 78.5 | 71.8 | 66.0 | 61.6 | 52.6 |
| **SNR** | 0.540 | 0.539 | 0.535 | 0.524 | 0.513 | 0.502 | 0.484 | 0.465 | 0.436 | 0.406 | 0.370 | 0.297 |

Table 7: **Classification accuracy and SNR on well-separated clusters across diffusion noise levels.**

| $\sigma_t$ | 0.010 | 0.026 | 0.044 | 0.076 | 0.108 | 0.140 | 0.205 |
|---|---|---|---|---|---|---|---|
| **Acc** | 73.53 | 74.64 | 75.17 | 75.54 | 75.83 | 75.93 | 76.10 |
| **SNR** | 0.0369 | 0.0372 | 0.0375 | 0.0380 | 0.0382 | 0.0384 | 0.0387 |

| $\sigma_t$ | 0.271 | 0.339 | 0.719 | 1.234 | 2.031 | 3.424 | 6.135 |
|---|---|---|---|---|---|---|---|
| **Acc** | 75.92 | 74.85 | 66.47 | 49.31 | 29.28 | 16.41 | 7.81 |
| **SNR** | 0.0384 | 0.0387 | 0.0350 | 0.0321 | 0.0282 | 0.0255 | 0.0211 |

Table 8: **Feature accuracy and SNR dynamics for DiT-XL/2 on `miniImageNet`.**

**Assets license information.** We primarily utilize the codebase of EDM [49], which is released under the Creative Commons Attribution-NonCommercial-ShareAlike 4.0 (CC BY-NC-SA 4.0) license. We also use code from the GitHub repository accompanying [19], which is licensed under the MIT License.

For the datasets, CIFAR datasets [50], ImageNet [51], Oxford 102 Flowers [93], and DTD [96] are publicly available for academic use. The CelebA dataset [97] is released for non-commercial research purposes only under a custom license. Oxford-IIIT Pet [91] is available under the CC BY-SA 4.0 license. The FFHQ dataset [98] is distributed by NVIDIA under the CC BY-NC-SA 4.0 license.

**Computational resources.** Most experiments are conducted on a single NVIDIA A40 GPU, except for training on subsets of images (e.g., Figure 8), which is performed using two A40 GPUs.

**Experimental details for Figure 1.**

- *Experimental details for Figure 1(a).* We train diffusion models based on the unified framework proposed by [49]. Specifically, we use the DDPM+ network, and use VP configuration for Figure 1(a). [49] has shown equivalence between VP configuration and the traditional DDPM setting, thus we call the models as DDPM* models. We train two models on CIFAR10 and CIFAR100, respectively. After training, we evaluate the learned representations via linear probing. For each noise level $\sigma(t)$, we corrupt the clean input $x_0$ with Gaussian noise to obtain $x_t = \sqrt{\bar{\alpha}_t}(x_0 + n)$, with $n \sim \mathcal{N}\left(0, \sigma_t^2 I\right)$ and $\sqrt{\bar{\alpha}_t} = 1/\sqrt{\sigma_t^2 + 1}$. We then extract features from the decoder's '16x16 block1' layer, which consistently yields the highest classification accuracy. A logistic regression classifier is trained on these features using the training split and evaluated on the test split. We perform the linear probe for each of the following noise levels: [0.002, 0.008, 0.023, 0.060, 0.140, 0.296, 0.585, 1.088, 1.923].

- *Experimental details for Figure 1(b).* We exactly follow the protocol in [19], using the same datasets which are subsets of CelebA [97, 99] and FFHQ [98], the same training procedure, and the same segmentation networks (MLPs). The only difference is that we use a newer latent diffusion model [3] pretrained on CelebAHQ from Hugging Face and the noise are added to the latent space. For feature extraction we concatenate the feature from the first layer of each resolution in the UNet's decoder (after upsampling them to the same resolution as the input). We perform segmentation for each of the following noise levels:[0.010, 0.015, 0.030, 0.053, 0.098, 0.189, 0.282, 0.379, 0.480, 0.766, 1.123].

**Experimental details for Figure 3 and Figure 15.** For the `MoLRG` experiments, we train the our parameterized model (4) following the setup provided in an open-source repository [100]. The model is trained on a $d = 5, n = 50, K = 3$ and $\delta = 0.2$ `MoLRG` dataset containing 12000 samples. Training is conducted for 200 epochs using DDPM scheduling with $T = 500$, employing the Adam optimizer

| Dataset | Classifier Type | $\sigma_t = 0.008$ | 0.023 | 0.060 | 0.140 | 0.296 | 0.585 | 1.088 | 1.923 |
|---------|----------------|---------|-------|-------|-------|-------|-------|-------|-------|
| CIFAR10 | Logistic Regression | 93.72 | 94.94 | 95.20 | 94.11 | 91.36 | 85.25 | 72.95 | 56.21 |
| CIFAR10 | Projection | 86.95 | 90.36 | 90.93 | 90.78 | 88.48 | 82.82 | 70.53 | 53.84 |

| Dataset | Classifier Type | $\sigma_t = 0.008$ | 0.023 | 0.060 | 0.140 | 0.296 | 0.585 | 1.088 | 1.923 |
|---------|----------------|---------|-------|-------|-------|-------|-------|-------|-------|
| TinyImageNet | Logistic Regression | 32.17 | 34.70 | 43.09 | 53.58 | 50.78 | 42.13 | 27.56 | 15.10 |
| TinyImageNet | Projection | 12.89 | 14.38 | 24.10 | 34.78 | 34.88 | 30.02 | 20.99 | 12.64 |

Table 9: **Logistic linear probing vs. projection-based classification accuracy on CIFAR10 (Top) and TinyImageNet (Bottom) across diffusion noise levels.**

with a learning rate of $5 \times 10^{-4}$. For SNR computation, we follow the definition in Section 3.3 since we have access to the ground-truth basis for the MoLRG data, i.e., $U_1^\star, U_2^\star$, and $U_3^\star \in \mathbb{R}^{50 \times 5}$. For probing we simply train a linear probe on the feature.

For both panels in Figure 3, we train our probe the same training set used for diffusion and test on five different MoLRG datasets with 9000 samples generated with five different random seeds, reporting the average accuracy and SNR at time steps [5, 10, 20, 40, 60, 80, 100, 120, 140, 160, 180, 240, 260, 280]. In Figure 15, we visualize the posterior estimations at time steps [5, 20, 60, 120, 260] by projecting them onto the union of the first columns of $U_1^\star, U_2^\star$, and $U_3^\star$ (a 3D space), then further projecting onto the 2D plane by a random $3 \times 2$ matrix with orthonormal columns. The subtitles of each visualization show the corresponding SNR calculated as explained above.

**Experimental details for Figure 5.** We use pre-trained EDM models [49] for CIFAR10 and ImageNet, extracting feature representations from the best-performing layer at each timestep. For the ImageNet model, features are extracted using images from classes in TinyImageNet [92]. Feature accuracy is evaluated via linear probing. To compute the SNR metric, we first normalize the extracted features by dividing each by its norm and subtracting the global mean. At each timestep, we perform class-wise SVD on the normalized features and compute SNR as defined in Section 3.3. Specifically, we use the top 5 right singular vectors of each class to form $U_k^\star$.

**Experimental details for Figure 6.** We use the DDPM++ network and VP configuration to train diffusion models[49] on the CIFAR10 dataset, using two network configurations: UNet-32 and UNet-64, by varying the embedding dimension of the UNet. Training dataset sizes range exponentially from $2^8$ to $2^{15}$. For each dataset size, both UNet-32 and UNet-64 are trained on the same subset of the training data. All models are trained with a duration of 50M images following the EDM training setup. After training, we calculate the generalization score as described in [53], using 10K generated images and the full training subset to compute the score.

**Experimental details for Figure 7 and Figure 8.** We use the DDPM++ architecture with the EDM configuration to train a UNet-128 diffusion model [49] on CIFAR10 and CIFAR100, using 4096 image training subsets. We track the evolution of representation dynamics throughout training. For Figure 7, FID [58] is computed using 50K generated samples compared against the full training dataset. Classification accuracy is obtained by extracting features from the training subset and evaluating a linear classifier on the full test set. Nearest neighbors are identified by computing the smallest $\ell_2$ distance between each generated image and the training subset.

**Experimental details for Figure 13.** We train individual DAEs using the DDPM++ network and VP configuration outlined in [49] at the following noise scales:

$$[0.002, 0.008, 0.023, 0.06, 0.14, 0.296, 0.585, 1.088, 1.923, 3.257].$$

Each model is trained for 500 epochs using the Adam optimizer [101] with a fixed learning rate of $1 \times 10^{-4}$. The sliced Wasserstein distance is computed according to the implementation described in [94].

**Experimental details for Table 1 and Table 2** For EDM, we use the official pre-trained checkpoints on ImageNet $64 \times 64$ from [49], and for DiT, we use the released DiT-XL/2 model pre-trained on

ImageNet $256 \times 256$ from [37]. As a baseline, we include the Hugging Face pre-trained MAE encoder (ViT-B/16) [95].

For diffusion models, features are extracted from the layer and timestep that achieve the highest probing accuracy, following [20]. After feature extraction, we adopt the probing protocol from [22], passing the extracted features through a probe consisting of a BatchNorm1d layer followed by a linear layer. To ensure fair comparisons, all input images are cropped or resized to $224 \times 224$, matching the resolution used for MAE training.

For ensembling, we extract features from two additional timesteps on either side of the optimal timestep. Independent probes are trained on these timesteps, yielding five probes in total. At test time, we apply a soft-voting ensemble by averaging the output logits from all five probes for the final prediction. Specifically, let $\boldsymbol{W}_t \in \mathbb{R}^{K \times d}$ be the linear classifier trained on features from timestep $t$, and let $\boldsymbol{h}_t \in \mathbb{R}^d$ denote the feature representation of a sample at timestep $t$. Considering neighboring timesteps $t - 2, t - 1, t + 1$, and $t + 2$, our ensemble prediction is computed as: $\hat{y} = \arg\max \left( \frac{1}{5} \sum_{t=t-2}^{t+2} \boldsymbol{W}_t \boldsymbol{h}_t \right)$.

We evaluate each method under varying levels of label noise, ranging from $0\%$ to $80\%$, by randomly mislabeling the specified percentage of training labels before applying linear probing. Performance is assessed on both the pre-training dataset and downstream transfer learning tasks. For pre-training evaluation, we use the images and classes from MiniImageNet [102] to reduce computational cost. For transfer learning, we evaluate on CIFAR100 [50], DTD [96], and Flowers102 [93].

# E    Proofs

We first provide some auxiliary results for proving Theorem 1.

## E.1    Groud truth posterior mean

We begin by deriving the ground truth posterior mean $\mathbb{E}\left[\boldsymbol{x}_0 | \boldsymbol{x}_t\right]$ under the MoLRG distribution. When $\boldsymbol{x}_0$ follows the noisy MoLRG assumption, the optimal solution $\hat{\boldsymbol{x}}_{\boldsymbol{\theta}}^{\star}(\boldsymbol{x}_t, t)$ to the training objective in Eq. (2) is exactly the posterior mean $\mathbb{E}\left[\boldsymbol{x}_0 | \boldsymbol{x}_t\right]$.

**Proposition 2.** *Suppose the data $\boldsymbol{x}_0$ is drawn from a noisy MoLRG data distribution with $K$-class and noise level $\delta$ introduced in Assumption 1. Then the optimal $\{U\}$ minimizing the loss (2) is the ground truth basis defined in (3), and the optimal DAE $\hat{\boldsymbol{x}}_{\boldsymbol{\theta}}^{\star}(\boldsymbol{x}_t, t)$ admits the analytical form:*

$$\hat{\boldsymbol{x}}_{\boldsymbol{\theta}}^{\star}(\boldsymbol{x}_t, t) = \sum_{l=1}^{K} w_l^{\star}(\boldsymbol{x}_t, t) \left( \zeta_t \boldsymbol{U}_l^{\star} \boldsymbol{U}_l^{\star\top} + \xi_t \widetilde{\boldsymbol{U}}_l^{\star} \widetilde{\boldsymbol{U}}_l^{\star\top} \right) \boldsymbol{x}_t, \tag{9}$$

*where $\zeta_t = \frac{1}{1+\sigma_t^2}$ and $\xi_t = \frac{\delta^2}{\delta^2+\sigma_t^2}$, and*

$$w_l^{\star}(\boldsymbol{x}_t, t) = \frac{\exp\left(g_l^{\star}(\boldsymbol{x}_t, t)\right)}{\sum_{s=1}^{K} \exp\left(g_s^{\star}(\boldsymbol{x}_t, t)\right)}, \quad g_l^{\star}(\boldsymbol{x}_t, t) = \frac{\zeta_t}{2\sigma_t^2} \|\boldsymbol{U}_l^{\star\top} \boldsymbol{x}_t\|^2 + \frac{\xi_t}{2\sigma_t^2} \|\widetilde{\boldsymbol{U}}_l^{\star\top} \boldsymbol{x}_t\|^2.$$

*Proof.* By [103] we see that the ground-truth score/posterior estimator calculated from the pdf is the global minimizer of the denoising score matching loss in (2), so here we first calculate the ground-truth score of the noisy MoLRG distribution and the corresponding posterior. We follow the same proof steps as in [32] Lemma 1 with a change of variable. Let $\boldsymbol{c}_k = \begin{bmatrix} \boldsymbol{a}_k \\ \boldsymbol{e}_k \end{bmatrix}$ and $\widehat{\boldsymbol{U}}_k = \begin{bmatrix} \boldsymbol{U}_k^{\star} & \delta\widetilde{\boldsymbol{U}}_k^{\star} \end{bmatrix}$ where $d$ and $D$ denote the dimensions of signal space and noise space as in definition, we first compute the conditional pdf

$$p_t(\boldsymbol{x}|Y = k)$$

$$= \int p_t\left(\boldsymbol{x}|Y = k, \boldsymbol{c}_k\right)\mathcal{N}(\boldsymbol{c}_k; \boldsymbol{0}, \boldsymbol{I}_{d+D}) \, \mathrm{d}\boldsymbol{c}_k$$

$$= \int p_t(\boldsymbol{x}|\boldsymbol{x}_0 = \widehat{\boldsymbol{U}}_k \boldsymbol{c}_k)\mathcal{N}\left(\boldsymbol{c}_k; \boldsymbol{0}, \boldsymbol{I}_{d+D}\right) \, \mathrm{d}\boldsymbol{c}_k$$

$$= \int \mathcal{N}(\boldsymbol{x}; s_t \widehat{\boldsymbol{U}}_k \boldsymbol{c}_k, \gamma_t^2 \boldsymbol{I}_n) \mathcal{N}(\boldsymbol{c}_k; \boldsymbol{0}, \boldsymbol{I}_{d+D}) \, \mathrm{d}\boldsymbol{c}_k$$

$$= \frac{1}{(2\pi)^{n/2}(2\pi)^{(d+D)/2}\gamma_t^n} \int \exp\left(-\frac{1}{2\gamma_t^2}\|\boldsymbol{x} - s_t \widehat{\boldsymbol{U}}_k \boldsymbol{c}_k\|^2\right) \exp\left(-\frac{1}{2}\|\boldsymbol{c}_k\|^2\right) \mathrm{d}\boldsymbol{c}_k$$

$$= \frac{1}{(2\pi)^{n/2}(2\pi)^{(d+D)/2}\gamma_t^n} \int \exp\left(-\frac{1}{2\gamma_t^2}\left(\boldsymbol{x}^\top \boldsymbol{x} - 2s_t \boldsymbol{x}^\top \widehat{\boldsymbol{U}}_k \boldsymbol{c}_k + s_t^2 \boldsymbol{c}_k^\top \widehat{\boldsymbol{U}}_k^\top \widehat{\boldsymbol{U}}_k \boldsymbol{c}_k + \gamma_t^2 \boldsymbol{c}_k^\top \boldsymbol{c}_k\right)\right) \mathrm{d}\boldsymbol{c}_k$$

$$= \frac{1}{(2\pi)^{n/2}\gamma_t^n} \left(\frac{s_t^2 + \gamma_t^2}{\gamma_t^2}\right)^{-d/2} \left(\frac{s_t^2\delta^2 + \gamma_t^2}{\gamma_t^2}\right)^{-D/2} \exp\left(-\frac{1}{2\gamma_t^2}\boldsymbol{x}^\top \left(\boldsymbol{I}_n - \frac{s_t^2}{s_t^2 + \gamma_t^2}\boldsymbol{U}_k^\star \boldsymbol{U}_k^{\star\top} - \frac{s_t^2\delta^2}{s_t^2\delta^2 + \gamma_t^2}\widetilde{\boldsymbol{U}}_k^\star \widetilde{\boldsymbol{U}}_k^{\star\top}\right) \boldsymbol{x}\right)$$

$$\int \frac{1}{(2\pi)^{d/2}} \left(\frac{\gamma_t^2}{s_t^2 + \gamma_t^2}\right)^{-d/2} \exp\left(-\frac{s_t^2 + \gamma_t^2}{2\gamma_t^2}\left\|\boldsymbol{a}_k - \frac{s_t}{s_t^2 + \gamma_t^2}\boldsymbol{U}_k^{\star\top}\boldsymbol{x}\right\|^2\right) \mathrm{d}\boldsymbol{a}_k$$

$$\int \frac{1}{(2\pi)^{D/2}} \left(\frac{\gamma_t^2}{s_t^2\delta^2 + \gamma_t^2}\right)^{-D/2} \exp\left(-\frac{s_t^2\delta^2 + \gamma_t^2}{2\gamma_t^2}\left\|\boldsymbol{e}_k - \frac{s_t\delta}{s_t^2\delta^2 + \gamma_t^2}\widetilde{\boldsymbol{U}}_k^{\star\top}\boldsymbol{x}\right\|^2\right) \mathrm{d}\boldsymbol{e}_k$$

$$= \frac{1}{(2\pi)^{n/2}} \frac{1}{(s_t^2 + \gamma_t^2)^{d/2}(s_t^2\delta^2 + \gamma_t^2)^{D/2}} \exp\left(-\frac{1}{2\gamma_t^2}\boldsymbol{x}^\top \left(\boldsymbol{I}_n - \frac{s_t^2}{s_t^2 + \gamma_t^2}\boldsymbol{U}_k^\star \boldsymbol{U}_k^{\star\top} - \frac{s_t^2\delta^2}{s_t^2\delta^2 + \gamma_t^2}\widetilde{\boldsymbol{U}}_k^\star \widetilde{\boldsymbol{U}}_k^{\star\top}\right) \boldsymbol{x}\right)$$

$$= \frac{1}{(2\pi)^{n/2} \det^{1/2}(s_t^2 \boldsymbol{U}_k^\star \boldsymbol{U}_k^{\star\top} + s_t^2\delta^2 \widetilde{\boldsymbol{U}}_k^\star \widetilde{\boldsymbol{U}}_k^{\star\top} + \gamma_t^2 \boldsymbol{I}_n)}$$
$$\exp\left(-\frac{1}{2}\boldsymbol{x}^T \left(s_t^2 \boldsymbol{U}_k^\star \boldsymbol{U}_k^{\star\top} + s_t^2\delta^2 \widetilde{\boldsymbol{U}}_k^\star \widetilde{\boldsymbol{U}}_k^{\star\top} + \gamma_t^2 \boldsymbol{I}_n\right)^{-1} \boldsymbol{x}\right)$$

$$= \mathcal{N}(\boldsymbol{x}; \boldsymbol{0}, s_t^2 \boldsymbol{U}_k^\star \boldsymbol{U}_k^{\star\top} + s_t^2\delta^2 \widetilde{\boldsymbol{U}}_k^\star \widetilde{\boldsymbol{U}}_k^{\star\top} + \gamma_t^2 \boldsymbol{I}_n),$$

where we repeatedly apply the pdf of multi-variate Gaussian and the second last equality uses $\det(s_t^2 \boldsymbol{U}_k^\star \boldsymbol{U}_k^{\star\top} + s_t^2\delta^2 \widetilde{\boldsymbol{U}}_k^\star \widetilde{\boldsymbol{U}}_k^{\star\top} + \gamma_t^2 \boldsymbol{I}_n) = (s_t^2 + \gamma_t^2)^d (s_t^2\delta^2 + \gamma_t^2)^D$ and $(s_t^2 \boldsymbol{U}_k^\star \boldsymbol{U}_k^{\star\top} + s_t^2\delta^2 \widetilde{\boldsymbol{U}}_k^\star \widetilde{\boldsymbol{U}}_k^{\star\top} + \gamma_t^2 \boldsymbol{I}_n)^{-1} = \left(\boldsymbol{I}_n - s_t^2/(s_t^2 + \gamma_t^2)\boldsymbol{U}_k^\star \boldsymbol{U}_k^{\star\top} - s_t^2\delta^2/(s_t^2\delta^2 + \gamma_t^2)\widetilde{\boldsymbol{U}}_k^\star \widetilde{\boldsymbol{U}}_k^{\star\top}\right)/\gamma_t^2$ because of the Woodbury matrix inversion lemma. Hence, with $\mathbb{P}(Y = k) = \pi_k$ for each $k \in [K]$, we have

$$p_t(\boldsymbol{x}) = \sum_{k=1}^K p_t(\boldsymbol{x}|Y = k)\mathbb{P}(Y = k) = \sum_{k=1}^K \pi_k \mathcal{N}(\boldsymbol{x}; \boldsymbol{0}, s_t^2 \boldsymbol{U}_k^\star \boldsymbol{U}_k^{\star\top} + s_t^2\delta^2 \widetilde{\boldsymbol{U}}_k^\star \widetilde{\boldsymbol{U}}_k^{\star\top} + \gamma_t^2 \boldsymbol{I}_n).$$

Now we can compute the score function

$$\nabla \log p_t(\boldsymbol{x}) = \frac{\nabla p_t(\boldsymbol{x})}{p_t(\boldsymbol{x})} = \frac{\sum_{k=1}^K \pi_k \mathcal{N}(\boldsymbol{x}; \boldsymbol{0}, s_t^2 \boldsymbol{U}_k^\star \boldsymbol{U}_k^{\star\top} + s_t^2\delta^2 \widetilde{\boldsymbol{U}}_k^\star \widetilde{\boldsymbol{U}}_k^{\star\top} + \gamma_t^2 \boldsymbol{I}_n) \left(-\frac{1}{\gamma_t^2}\boldsymbol{x} + \frac{s_t^2}{\gamma_t^2(s_t^2 + \gamma_t^2)}\boldsymbol{U}_k^\star \boldsymbol{U}_k^{\star\top}\boldsymbol{x} + \frac{s_t^2\delta^2}{\gamma_t^2(s_t^2\delta^2 + \gamma_t^2)}\widetilde{\boldsymbol{U}}_k^\star \widetilde{\boldsymbol{U}}_k^{\star T}\boldsymbol{x}\right)}{\sum_{k=1}^K \pi_k \mathcal{N}(\boldsymbol{x}; \boldsymbol{0}, s_t^2 \boldsymbol{U}_k^\star \boldsymbol{U}_k^{\star\top} + s_t^2\delta^2 \widetilde{\boldsymbol{U}}_k^\star \widetilde{\boldsymbol{U}}_k^{\star\top} + \gamma_t^2 \boldsymbol{I}_n)}$$

$$= -\frac{1}{\gamma_t^2}\left(\boldsymbol{x} - \frac{\sum_{k=1}^K \pi_k \mathcal{N}(\boldsymbol{x}; \boldsymbol{0}, s_t^2 \boldsymbol{U}_k \boldsymbol{U}_k^{\star\top} + s_t^2\delta^2 \widetilde{\boldsymbol{U}}_k^\star \widetilde{\boldsymbol{U}}_k^{\star\top} + \gamma_t^2 \boldsymbol{I}_n)\left(\frac{s_t^2}{s_t^2 + \gamma_t^2}\boldsymbol{U}_k^\star \boldsymbol{U}_k^{\star\top}\boldsymbol{x}) + \frac{s_t^2\delta^2}{s_t^2\delta^2 + \gamma_t^2}\widetilde{\boldsymbol{U}}_k^\star \widetilde{\boldsymbol{U}}_k^{\star\top}\boldsymbol{x})\right)}{\sum_{k=1}^K \pi_k \mathcal{N}(\boldsymbol{x}; \boldsymbol{0}, s_t^2 \boldsymbol{U}_k^\star \boldsymbol{U}_k^{\star\top} + s_t^2\delta^2 \widetilde{\boldsymbol{U}}_k^\star \widetilde{\boldsymbol{U}}_k^{\star\top} + \gamma_t^2 \boldsymbol{I}_n)}\right).$$

According to Tweedie's formula, we have

$$\mathbb{E}[\boldsymbol{x}_0|\boldsymbol{x}_t] = \frac{\boldsymbol{x}_t + \gamma_t^2 \nabla \log p_t(\boldsymbol{x}_t)}{s_t}$$

$$= \frac{s_t}{s_t^2 + \gamma_t^2} \frac{\sum_{k=1}^K \pi_k \mathcal{N}(\boldsymbol{x}; \boldsymbol{0}, s_t^2 \boldsymbol{U}_k^\star \boldsymbol{U}_k^{\star\top} + s_t^2\delta^2 \widetilde{\boldsymbol{U}}_k^\star \widetilde{\boldsymbol{U}}_k^{\star\top} + \gamma_t^2 \boldsymbol{I}_n)\boldsymbol{U}_k^\star \boldsymbol{U}_k^{\star\top}\boldsymbol{x}}{\mathcal{N}(\boldsymbol{x}; \boldsymbol{0}, s_t^2 \boldsymbol{U}_k^\star \boldsymbol{U}_k^{\star\top} + s_t^2\delta^2 \widetilde{\boldsymbol{U}}_k^\star \widetilde{\boldsymbol{U}}_k^{\star\top} + \gamma_t^2 \boldsymbol{I}_n)}$$

$$+ \frac{s_t\delta^2}{s_t^2\delta^2 + \gamma_t^2} \frac{\sum_{k=1}^K \pi_k \mathcal{N}(\boldsymbol{x}; \boldsymbol{0}, s_t^2 \boldsymbol{U}_k^\star \boldsymbol{U}_k^{\star\top} + s_t^2\delta^2 \widetilde{\boldsymbol{U}}_k^\star \widetilde{\boldsymbol{U}}_k^{\star\top} + \gamma_t^2 \boldsymbol{I}_n)\widetilde{\boldsymbol{U}}_k^\star \widetilde{\boldsymbol{U}}_k^{\star\top}\boldsymbol{x}}{\mathcal{N}(\boldsymbol{x}; \boldsymbol{0}, s_t^2 \boldsymbol{U}_k^\star \boldsymbol{U}_k^{\star\top} + s_t^2\delta^2 \widetilde{\boldsymbol{U}}_k^\star \widetilde{\boldsymbol{U}}_k^{\star\top} + \gamma_t^2 \boldsymbol{I}_n)}$$

$$= \frac{s_t}{s_t^2 + \gamma_t^2} \frac{\sum_{k=1}^{K} \pi_k \exp\left(\phi_t \|\boldsymbol{U}_k^{\star\top}\boldsymbol{x}_t\|^2\right) \exp\left(\psi_t \|\widetilde{\boldsymbol{U}}_k^{\star\top}\boldsymbol{x}_t\|^2\right) \boldsymbol{U}_k^{\star}\boldsymbol{U}_k^{\star\top}\boldsymbol{x}_t}{\sum_{k=1}^{K} \pi_k \exp\left(\phi_t \|\boldsymbol{U}_k^{\star\top}\boldsymbol{x}_t\|^2\right) \exp\left(\psi_t \|\widetilde{\boldsymbol{U}}_k^{\star\top}\boldsymbol{x}_t\|^2\right)}$$

$$+ \frac{s_t \delta^2}{s_t^2 \delta^2 + \gamma_t^2} \frac{\sum_{k=1}^{K} \pi_k \exp\left(\phi_t \|\boldsymbol{U}_k^{\star\top}\boldsymbol{x}_t\|^2\right) \exp\left(\psi_t \|\widetilde{\boldsymbol{U}}_k^{\star\top}\boldsymbol{x}_t\|^2\right) \widetilde{\boldsymbol{U}}_k^{\top}\widetilde{\boldsymbol{U}}_k^{\star\top}\boldsymbol{x}_t}{\sum_{k=1}^{K} \pi_k \exp\left(\phi_t \|\boldsymbol{U}_k^{\star\top}\boldsymbol{x}_t\|^2\right) \exp\left(\psi_t \|\widetilde{\boldsymbol{U}}_k^{\star\top}\boldsymbol{x}_t\|^2\right)},$$

with $\phi_t = s_t^2/(2\gamma_t^2(s_t^2 + \gamma_t^2))$ and $\psi_t = s_t^2 \delta^2/(2\gamma_t^2(s_t^2 \delta^2 + \gamma_t^2))$. The final equality uses the pdf of multi-variant Gaussian and the matrix inversion lemma discussed earlier. Since $\pi_k$ is consistent for all $k$ and $s_t = 1$, we have

$$\mathbb{E}\left[\boldsymbol{x}_0|\boldsymbol{x}_t\right] = \sum_{k=1}^{K} w_k^{\star}(\boldsymbol{x}_t) \left(\frac{1}{1+\sigma_t^2}\boldsymbol{U}_k^{\star}\boldsymbol{U}_k^{\star\top} + \frac{\delta^2}{\delta^2 + \sigma_t^2}\widetilde{\boldsymbol{U}}_k^{\star}\widetilde{\boldsymbol{U}}_k^{\star\top}\right)\boldsymbol{x}_t$$

$$\text{where } w_k^{\star}(\boldsymbol{x}_t) := \frac{\exp\left(\frac{1}{2\sigma_t^2(1+\sigma_t^2)}\|\boldsymbol{U}_k^{\star\top}\boldsymbol{x}_t\|^2 + \frac{\delta^2}{2\sigma_t^2(\delta^2+\sigma_t^2)}\|\widetilde{\boldsymbol{U}}_k^{\star\top}\boldsymbol{x}_t\|^2\right)}{\sum_{k=1}^{K}\exp\left(\frac{1}{2\sigma_t^2(1+\sigma_t^2)}\|\boldsymbol{U}_k^{\star\top}\boldsymbol{x}_t\|^2 + \frac{\delta^2}{2\sigma_t^2(\delta^2+\sigma_t^2)}\|\widetilde{\boldsymbol{U}}_k^{\star\top}\boldsymbol{x}_t\|^2\right)}.$$

Finally we show the equivalence between ground-truth posterior and our parameterized DAE when its weights are just the ground truth basis. We begin by rewriting the optimal posterior function by leveraging the fact that $\widetilde{\boldsymbol{U}}_l^{\star} := \begin{bmatrix} \boldsymbol{U}_1^{\star} & \cdots & \boldsymbol{U}_{l-1}^{\star} & \boldsymbol{U}_{l+1}^{\star} & \cdots \boldsymbol{U}_K^{\star} \end{bmatrix}$

$$\mathbb{E}\left[\boldsymbol{x}_0|\boldsymbol{x}_t\right] = \sum_{l=1}^{K} w_l^{\star}(\boldsymbol{x}_t, t) \left(\zeta_t \boldsymbol{U}_l^{\star}\boldsymbol{U}_l^{\star\top} + \xi_t \widetilde{\boldsymbol{U}}_l^{\star}\widetilde{\boldsymbol{U}}_l^{\star\top}\right)\boldsymbol{x}_t$$

$$= \sum_{l=1}^{K} w_l^{\star}(\boldsymbol{x}_t, t) \left(\zeta_t \boldsymbol{U}_l^{\star}\boldsymbol{U}_l^{\star\top} + \xi_t \sum_{j\neq l} \boldsymbol{U}_j^{\star}\boldsymbol{U}_j^{\star\top}\right)\boldsymbol{x}_t$$

$$= \sum_{l=1}^{K} w_l^{\star}(\boldsymbol{x}_t, t) \left(\zeta_t \boldsymbol{U}_l^{\star}\boldsymbol{U}_l^{\star\top}\right) + \sum_{l=1}^{K} (w_l^{\star}(\boldsymbol{x}_t, t)\xi_t \sum_{j\neq l} \boldsymbol{U}_j^{\star}\boldsymbol{U}_j^{\star\top})\boldsymbol{x}_t$$

$$= \sum_{l=1}^{K} \left(w_l^{\star}(\boldsymbol{x}_t, t)\zeta_t \boldsymbol{U}_l^{\star}\boldsymbol{U}_l^{\star\top}\boldsymbol{x}_t + \xi_t \sum_{j\neq l} w_j^{\star}(\boldsymbol{x}_t, t)\boldsymbol{U}_l^{\star}\boldsymbol{U}_l^{\star\top}\boldsymbol{x}_t\right)$$

$$= \sum_{l=1}^{K} \left(w_l^{\star}(\boldsymbol{x}_t, t)\zeta_t \boldsymbol{U}_l^{\star}\boldsymbol{U}_l^{\star\top}\boldsymbol{x}_t + \xi_t\left(1 - w_l^{\star}(\boldsymbol{x}_t, t)\right)\boldsymbol{U}_l^{\star}\boldsymbol{U}_l^{\star\top}\boldsymbol{x}_t\right)$$

$$= \sum_{l=1}^{K} \left(\zeta_t w_l^{\star}(\boldsymbol{x}_t, t) + \xi_t(1 - w_l^{\star}(\boldsymbol{x}_t, t))\right)\boldsymbol{U}_l^{\star}\boldsymbol{U}_l^{\star\top}\boldsymbol{x}_t$$

$$= \sum_{l=1}^{K} \left[\xi_t + (\zeta_t - \xi_t)w_l^{\star}(\boldsymbol{x}_t, t)\right]\boldsymbol{U}_l^{\star}\boldsymbol{U}_l^{\star\top}\boldsymbol{x}_t.$$

Now if we let $\boldsymbol{U}^{\star} = \begin{bmatrix} \boldsymbol{U}_1 & \boldsymbol{U}_2 & \dots & \boldsymbol{U}_K \end{bmatrix} \in \mathcal{O}^{n \times Kd}$ and $\boldsymbol{D}^{\star}(\boldsymbol{x}_t, t) = \text{diag}\left(\beta_1^{\star}\boldsymbol{I}_d, \dots, \beta_K^{\star}\boldsymbol{I}_d\right)$ to be a block-diagonal matrix. Each $\beta_l^{\star}$ is defined as $\beta_l^{\star} = \xi_t + (\zeta_t - \xi_t)w_l^{\star}(\boldsymbol{x}_t, t)$, we can then write:

$$\mathbb{E}\left[\boldsymbol{x}_0|\boldsymbol{x}_t\right] = \boldsymbol{U}^{\star}\boldsymbol{D}^{\star}(\boldsymbol{x}_t, t)\boldsymbol{U}^{\star T}\boldsymbol{x}_t.$$

Thus, the optimal solution for our network parametrization as defined in (4) is exactly $\mathbb{E}\left[\boldsymbol{x}_0|\boldsymbol{x}_t\right]$. And by such equivalence, the optimality of the DAE is induced. $\square$

### E.2  Proof of Theorem 1

We first state the formal version of Theorem 1.

To simplify the calculation of SNR as introduced in Section 3.3 on feature representations, which involves the expectation over the softmax term $w_k^\star$, we approximate $\hat{\boldsymbol{x}}_{\boldsymbol{\theta}}^\star$ as follows:

$$\hat{\boldsymbol{x}}_{approx}^\star(\boldsymbol{x}, t) = \sum_{k=1}^{K} \hat{w}_k^\star \left( \zeta_t \boldsymbol{U}_k^\star \boldsymbol{U}_k^{\star\top} + \xi_t \widetilde{\boldsymbol{U}}_k^\star \widetilde{\boldsymbol{U}}_k^{\star\top} \right) \boldsymbol{x},$$

$$\text{where } \hat{w}_k^\star := \frac{\exp\left(\mathbb{E}_{\boldsymbol{x}_t}[g_k^\star(\boldsymbol{x}_t, t)]\right)}{\sum_{k=1}^{K} \exp\left(\mathbb{E}_{\boldsymbol{x}_t}[g_k^\star(\boldsymbol{x}_t, t)]\right)}, \ \zeta_t = \frac{1}{1+\sigma_t^2}, \ \xi_t = \frac{\delta^2}{\delta^2 + \sigma_t^2}, \quad (10)$$

$$\text{and } g_k^\star(\boldsymbol{x}, t) = \frac{\zeta_t}{2\sigma_t^2} \|\boldsymbol{U}_k^{\star\top} \boldsymbol{x}\|^2 + \frac{\xi_t}{2\sigma_t^2} \|\widetilde{\boldsymbol{U}}_k^{\star\top} \boldsymbol{x}\|^2.$$

In other words, we use $\hat{w}_k^\star$ in (10) to approximate $w_k^\star(\boldsymbol{x}_t, t)$ in Proposition 1 by taking expectation inside the softmax with respect to $\boldsymbol{x}_t$. This allows us to treat $\hat{w}_k^\star$ as a constant when calculating SNR, making the analysis more tractable while maintaining $\mathbb{E}[\|\boldsymbol{U}_l^\star \hat{\boldsymbol{h}}_{\boldsymbol{\theta}}^\star(\boldsymbol{x}_t, t)\|^2] \approx \mathbb{E}[\|\boldsymbol{U}_l^\star \hat{\boldsymbol{h}}_{approx}^\star(\boldsymbol{x}_0, t)\|^2]$ for all $l \in [K]$. We verify the tightness of this approximation at Appendix C (Figure 14). With this approximation, we state the theorem as follows:

**Theorem 2.** *Let data $\boldsymbol{x}_0$ be any arbitrary data point drawn from the* MoLRG *distribution defined in Assumption 1 and let $k$ denote the true class $\boldsymbol{x}_0$ belongs to. Then* SNR *introduced in Section 3.3 depends on the noise level $\sigma_t$ in the following form:*

$$\text{SNR}(\hat{\boldsymbol{x}}_{approx}^\star, t) = \frac{1+\sigma_t^2}{(K-1)(\delta^2 + \sigma_t^2)} \cdot \left( \frac{1 + \frac{\sigma_t^2}{\delta^2} h(\hat{w}_k^\star, \delta)}{1 + \frac{\sigma_t^2}{\delta^2} h(\hat{w}_l^\star, \delta)} \right)^2 \quad (11)$$

*where $h(w, \delta) := (1 - \delta^2)w + \delta^2$. Since $\delta$ is fixed, $h(w, \delta)$ is a monotonically increasing function with respect to $w$. Note that here $\delta$ represents the magnitude of the fixed intrinsic noise in the data where $\sigma_t$ denotes the level of additive Gaussian noise introduced during the diffusion training process.*

*Proof.* Following the definition of SNR as defined in Section 3.3, Lemma 2 and the fact that $k \sim \text{Mult}(K, \pi_k)$ with $\pi_1 = \cdots = \pi_K = 1/K$, we can write

$$\text{SNR}(\hat{\boldsymbol{x}}_{approx}^\star, t) = \frac{\mathbb{E}_{\boldsymbol{x}_t}[\|\boldsymbol{U}_k^\star \boldsymbol{U}_k^{\star\top} \hat{\boldsymbol{x}}_{approx}^\star(\boldsymbol{x}_t, t)\|^2]}{\mathbb{E}_{\boldsymbol{x}_t}[\sum_{l \neq k} \|\boldsymbol{U}_l^\star \boldsymbol{U}_l^{\star\top} \hat{\boldsymbol{x}}_{approx}^\star(\boldsymbol{x}_t, t)\|^2]} = \frac{\mathbb{E}_{\boldsymbol{x}_t}[\|\boldsymbol{U}_k^\star \boldsymbol{U}_k^{\star\top} \hat{\boldsymbol{x}}_{approx}^\star(\boldsymbol{x}_t, t)\|^2]}{\sum_{l \neq k} \mathbb{E}_{\boldsymbol{x}_t}[\|\boldsymbol{U}_l^\star \boldsymbol{U}_l^{\star\top} \hat{\boldsymbol{x}}_{approx}^\star(\boldsymbol{x}_t, t)\|^2]}$$

$$= \frac{\left( \frac{\hat{w}_k^\star}{1+\sigma_t^2} + \frac{(K-1)\delta^2 \hat{w}_l^\star}{\delta^2 + \sigma_t^2} \right)^2 (1 + \sigma_t^2) d}{(K-1) \left( \frac{\hat{w}_l^\star}{1+\sigma_t^2} + \frac{\delta^2(\hat{w}_k^\star + (K-2)\hat{w}_l^\star)}{\delta^2 + \sigma_t^2} \right)^2 (\delta^2 + \sigma_t^2) d}$$

$$= \frac{1 + \sigma_t^2}{(K-1)(\delta^2 + \sigma_t^2)} \cdot \left( \frac{\hat{w}_k^\star \delta^2 + \hat{w}_k^\star \sigma_t^2 + (K-1)\delta^2 \hat{w}_l^\star + (K-1)\delta^2 \hat{w}_l^\star \sigma_t^2}{\hat{w}_l^\star \delta^2 + \hat{w}_l^\star \sigma_t^2 + \delta^2 \hat{w}_k^\star + (K-2)\delta^2 \hat{w}_l^\star + \delta^2 \hat{w}_k^\star \sigma_t^2 + (K-2)\delta^2 \hat{w}_l^\star \sigma_t^2} \right)^2$$

$$= \frac{1 + \sigma_t^2}{(K-1)(\delta^2 + \sigma_t^2)} \cdot \left( \frac{\delta^2 + \sigma_t^2 \left( \hat{w}_k^\star + (K-1)\delta^2 \hat{w}_l^\star \right)}{\delta^2 + \sigma_t^2 \left( \hat{w}_l^\star + \delta^2 \hat{w}_k^\star + (K-2)\delta^2 \hat{w}_l^\star \right)} \right)^2$$

$$= \frac{1 + \sigma_t^2}{(K-1)(\delta^2 + \sigma_t^2)} \cdot \left( \frac{1 + \frac{\sigma_t^2}{\delta^2} \left( (1-\delta^2)\hat{w}_k^\star + \delta^2(\hat{w}_k^\star + (K-1)\hat{w}_l^\star) \right)}{1 + \frac{\sigma_t^2}{\delta^2} \left( (1-\delta^2)\hat{w}_l^\star + \delta^2(\hat{w}_l^\star + \hat{w}_k^\star + (K-2)\hat{w}_l^\star) \right)} \right)^2$$

$$= \frac{1 + \sigma_t^2}{(K-1)(\delta^2 + \sigma_t^2)} \cdot \left( \frac{1 + \frac{\sigma_t^2}{\delta^2} \left( (1-\delta^2)\hat{w}_k^\star + \delta^2 \right)}{1 + \frac{\sigma_t^2}{\delta^2} \left( (1-\delta^2)\hat{w}_l^\star + \delta^2 \right)} \right)^2$$

$$= \frac{1 + \sigma_t^2}{(K-1)(\delta^2 + \sigma_t^2)} \cdot \left( \frac{1 + \frac{\sigma_t^2}{\delta^2} h(\hat{w}_k^\star, \delta)}{1 + \frac{\sigma_t^2}{\delta^2} h(\hat{w}_l^\star, \delta)} \right)^2.$$

where $h(w, \delta) := (1 - \delta^2)w + \delta^2$, and we set $C_t = \frac{1+\sigma_t^2}{(\delta^2 + \sigma_t^2)}$. $\qquad\square$

**Lemma 1.** *Let $g_1, g_2, \ldots, g_K$ be a sequence of real-valued inputs, and fix an index $k \in [K]$. Assume that $g_l = g_j$ for all $l, j \neq k$; that is, all entries except $g_k$ share the same value. Let the softmax weights be defined as*

$$w_j = \frac{\exp(g_j)}{\sum_{j=1}^{K} \exp(g_j)},$$

*Then we have:*

$$w_k = \frac{\exp(g_k - g_l)}{K - 1 + \exp(g_k - g_l)} \quad \text{and} \quad w_l = \frac{1}{K - 1 + \exp(g_k - g_l)} \quad \text{for all } l \neq k.$$

*Proof.* The softmax weight for class $k$ is:

$$w_k = \frac{\exp(g_k)}{\sum_{j=1}^{K} \exp(g_j)}.$$

We simplify the denominator by factoring out $\exp(g_l)$:

$$\sum_{j=1}^{K} \exp(g_j) = \exp(g_k) + \sum_{j \neq k} \exp(g_l) = \exp(g_k) + (K - 1) \exp(g_l).$$

Therefore,

$$w_k = \frac{\exp(g_k)}{\exp(g_k) + (K-1)\exp(g_l)} = \frac{1}{1 + (K-1)\exp(g_l - g_k)} = \frac{\exp(g_k - g_l)}{(K-1) + \exp(g_k - g_l)}.$$

For any $l \neq k$, we similarly have:

$$w_l = \frac{\exp(g_l)}{\exp(g_k) + (K-1)\exp(g_l)} = \frac{1}{(K-1) + \exp(g_k - g_l)}.$$

This simple fact can also be proven by applying the log-sum-exp trick. $\qquad \square$

**Lemma 2.** *Given the setup of a $K$-class* `MoLRG` *data distribution defined in (3), consider the following approximate posterior mean function:*

$$\hat{\boldsymbol{x}}_{approx}^{\star}(\boldsymbol{x}, t) = \sum_{k=1}^{K} \hat{w}_k^{\star} \left( \zeta_t \boldsymbol{U}_k^{\star} \boldsymbol{U}_k^{\star \top} + \xi_t \widetilde{\boldsymbol{U}}_k^{\star} \widetilde{\boldsymbol{U}}_k^{\star \top} \right) \boldsymbol{x},$$

$$\text{where } \hat{w}_k^{\star} := \frac{\exp\left(\mathbb{E}_{\boldsymbol{x}_t}[g_k^{\star}(\boldsymbol{x}_t, t)]\right)}{\sum_{k=1}^{K} \exp\left(\mathbb{E}_{\boldsymbol{x}_t}[g_k^{\star}(\boldsymbol{x}_t, t)]\right)}, \ \zeta_t = \frac{1}{1 + \sigma_t^2}, \ \xi_t = \frac{\delta^2}{\delta^2 + \sigma_t^2},$$

$$\text{and } g_k^{\star}(\boldsymbol{x}, t) = \frac{\zeta_t}{2\sigma_t^2} \|\boldsymbol{U}_k^{\star \top} \boldsymbol{x}\|^2 + \frac{\xi_t}{2\sigma_t^2} \|\widetilde{\boldsymbol{U}}_k^{\star \top} \boldsymbol{x}\|^2.$$

*That is, we consider a simplified form of the expected posterior mean from Proposition 1, where the expectation is taken inside the softmax argument (i.e., over $g_k^{\star}(\boldsymbol{x}_t, t)$) to obtain tractable approximate weights $\hat{w}_k^{\star}$.*

*Under this approximation, for any sample $\boldsymbol{x}_0$ from class $k$, i.e., $\boldsymbol{x}_0 = \boldsymbol{U}_k^{\star} \boldsymbol{a}_i + \delta \widetilde{\boldsymbol{U}}_k^{\star} \boldsymbol{e}_i$, we have:*

$$\mathbb{E}_{\boldsymbol{x}_t}[\|\boldsymbol{U}_k^{\star} \boldsymbol{U}_k^{\star \top} \hat{\boldsymbol{x}}_{approx}^{\star}(\boldsymbol{x}_t, t)\|^2] = \left( \frac{\hat{w}_k^{\star}}{1 + \sigma_t^2} + \frac{(K-1)\delta^2 \hat{w}_l^{\star}}{\delta^2 + \sigma_t^2} \right)^2 (d + \sigma_t^2 d) \tag{12}$$

$$\mathbb{E}_{\boldsymbol{x}_t}[\|\boldsymbol{U}_l^{\star} \boldsymbol{U}_l^{\star \top} \hat{\boldsymbol{x}}_{approx}^{\star}(\boldsymbol{x}_t, t)\|^2] = \left( \frac{\hat{w}_l^{\star}}{1 + \sigma_t^2} + \frac{\delta^2 (\hat{w}_k + (K-2)\hat{w}_l^{\star})}{\delta^2 + \sigma_t^2} \right)^2 (\delta^2 d + \sigma_t^2 d) \tag{13}$$

$$\mathbb{E}_{\boldsymbol{x}_t}[\|\hat{\boldsymbol{x}}_{approx}^{\star}(\boldsymbol{x}_t,t)\|^2] = \underbrace{\left(\frac{\hat{w}_k^{\star}}{1+\sigma_t^2} + \frac{(K-1)\delta^2\hat{w}_l^{\star}}{\delta^2+\sigma_t^2}\right)^2 (d+\sigma_t^2 d)}_{\mathbb{E}\,\|\boldsymbol{U}_k^{\star}\boldsymbol{U}_k^{\star\top}\hat{\boldsymbol{x}}_{approx}^{\star}(\boldsymbol{x}_t,t)\|^2}$$

$$+ \underbrace{(K-1)\left(\frac{\hat{w}_l^{\star}}{1+\sigma_t^2} + \frac{\delta^2(\hat{w}_k^{\star}+(K-2)\hat{w}_l^{\star})}{\delta^2+\sigma_t^2}\right)^2 (\delta^2 d + \sigma_t^2 d)}_{\mathbb{E}[\sum_{l\neq k}^{K}\boldsymbol{U}_l^{\star}\boldsymbol{U}_l^{\star\top}\hat{\boldsymbol{x}}_{approx}^{\star}(\boldsymbol{x}_t,t)\|^2]} \tag{14}$$

*and*

$$\hat{w}_k^{\star} = \frac{\exp\left(D_t\right)}{K-1+\exp\left(D_t\right)},$$
$$\hat{w}_l^{\star} = \frac{1}{K-1+\exp\left(D_t\right)}. \tag{15}$$

*for all class index $l \neq k$, where $D_t = \frac{(1-\delta^2)d}{2\sigma_t^2(1+\sigma_t^2)}$.*

*Proof.* Throughout the proof, we use the following notation for slices of vectors.

$$\boldsymbol{e}_i[a:b] \qquad \text{Slices of vector } \boldsymbol{e}_i \text{ from } a\text{th entry to } b\text{th entry.}$$

We begin with the softmax terms. Since each class has its unique disjoint subspace, it suffices to consider $g_k(\boldsymbol{x}_0,t)$ and $g_l(\boldsymbol{x}_0,t)$ for any $l \neq k$. Let $a_t = \frac{1}{2\sigma_t^2(1+\sigma_t^2)}$ and $c_t = \frac{\delta^2}{2\sigma_t^2(\delta^2+\sigma_t^2)}$, we have:

$$\mathbb{E}[g_k^{\star}(\boldsymbol{x}_t,t)] = \mathbb{E}[a_t\|\boldsymbol{U}_k^{\star\top}\boldsymbol{x}_t\|^2 + c_t\|\widetilde{\boldsymbol{U}}_k^{\star\top}\boldsymbol{x}_t\|^2]$$
$$= \mathbb{E}[a_t\|\boldsymbol{U}_k^{\star\top}(\boldsymbol{U}_k^{\star}\boldsymbol{a}_i + \delta\widetilde{\boldsymbol{U}}_k^{\star\top}\boldsymbol{e}_i + \sigma_t\boldsymbol{\epsilon}_i)\|^2] + \mathbb{E}[c_t\|\widetilde{\boldsymbol{U}}_k^{\star\top}(\boldsymbol{U}_k^{\star}\boldsymbol{a}_i + \delta\widetilde{\boldsymbol{U}}_k^{\star\top}\boldsymbol{e}_i + \sigma_t\boldsymbol{\epsilon}_i)\|^2]$$
$$= \mathbb{E}[a_t\|\boldsymbol{a}_i + \sigma_t\boldsymbol{U}_k^{\star\top}\boldsymbol{\epsilon}_i\|^2] + \mathbb{E}[c_t\|\delta\boldsymbol{e}_i + \sigma_t\widetilde{\boldsymbol{U}}_k^{\star\top}\boldsymbol{\epsilon}_i\|^2]$$
$$= a_t(d + \sigma_t^2 d) + c_t\left(\delta^2(K-1)d + \sigma_t^2(K-1)d\right).$$

where the last equality follows from $\boldsymbol{a}_i \overset{i.i.d.}{\sim} \mathcal{N}(\boldsymbol{0}, \boldsymbol{I}_d)$, $\boldsymbol{e}_i \overset{i.i.d.}{\sim} \mathcal{N}(\boldsymbol{0}, \boldsymbol{I}_{(K-1)d})$ and $\boldsymbol{\epsilon}_i \overset{i.i.d.}{\sim} \mathcal{N}(\boldsymbol{0}, \boldsymbol{I}_n)$.
Without loss of generality, assume the $j = k+1$, we have:

$$\mathbb{E}[g_l^{\star}(\boldsymbol{x}_t,t)] = \mathbb{E}[a_t\|\boldsymbol{U}_l^{\star\top}\boldsymbol{x}_t\|^2 + c_t\|\widetilde{\boldsymbol{U}}_l^{\star\top}\boldsymbol{x}_t\|^2]$$
$$= \mathbb{E}[a_t\|\boldsymbol{U}_l^{\star\top}(\boldsymbol{U}_k^{\star}\boldsymbol{a}_i + \delta\widetilde{\boldsymbol{U}}_k^{\star}\boldsymbol{e}_i + \sigma_t\boldsymbol{\epsilon}_i)\|^2] + \mathbb{E}[c_t\|\widetilde{\boldsymbol{U}}_l^{\perp\top}(\boldsymbol{U}_k^{\star}\boldsymbol{a}_i + \delta\widetilde{\boldsymbol{U}}_k^{\star}\boldsymbol{e}_i + \sigma_t\boldsymbol{\epsilon}_i)\|^2]$$
$$= \mathbb{E}[a_t\|\delta\boldsymbol{e}_i[1:d] + \sigma_t\widetilde{\boldsymbol{U}}_l^{\star\top}\boldsymbol{\epsilon}_i\|^2] + \mathbb{E}\left[c_t\left\|\begin{bmatrix}\boldsymbol{a}_i \\ \boldsymbol{0}\in\mathbb{R}^{D-d}\end{bmatrix} + \delta\begin{bmatrix}\boldsymbol{0}\in\mathbb{R}^d \\ \boldsymbol{e}_i[d:D]]\end{bmatrix} + \sigma_t\widetilde{\boldsymbol{U}}_l^{\star\top}\boldsymbol{\epsilon}_i\right\|^2\right]$$
$$= a_t\left(\delta^2 d + \sigma_t^2 d\right) + c_t\left(d + \delta^2(K-2)d + \sigma_t^2(K-1)d\right).$$

Hence we have

$$\mathbb{E}[g_k^{\star}(\boldsymbol{x}_t,t)] - \mathbb{E}[g_l^{\star}(\boldsymbol{x}_t,t)]$$
$$= a_t\left(d + \sigma_t^2 d - \delta^2 d - \sigma_t^2 d\right) + c_t\left(\delta^2(K-1)d + \sigma_t^2(K-1)d - d - \delta^2(K-2)d - \sigma_t^2(K-1)d\right)$$
$$= a_t(1-\delta^2)d + c_t(\delta^2-1)d$$
$$= \frac{(1-\delta^2)d}{2\sigma_t^2(1+\sigma_t^2)} + \frac{\delta^2 d(\delta^2-1)}{2\sigma_t^2(\delta^2+\sigma_t^2)} = \frac{(\delta^2-1)^2 d}{2(1+\sigma_t^2)(\delta^2+\sigma_t^2)}.$$

This, together with Lemma 1, yield (15).

Now we prove (12):

$$\boldsymbol{U}_k^{\star}\boldsymbol{U}_k^{\star\top}\hat{\boldsymbol{x}}_{approx}^{\star}(\boldsymbol{x}_t,t) = \hat{w}_k^{\star}\boldsymbol{U}_k^{\star}\boldsymbol{U}_k^{\star\top}\left(\frac{1}{1+\sigma_t^2}\boldsymbol{U}_k^{\star}\boldsymbol{U}_k^{\star\top} + \frac{\delta^2}{\delta^2+\sigma_t^2}\widetilde{\boldsymbol{U}}_k^{\star}\widetilde{\boldsymbol{U}}_k^{\star\top}\right)\boldsymbol{x}_t$$

$$+ \sum_{l \neq k} \hat{w}_l^\star \boldsymbol{U}_k^\star \boldsymbol{U}_k^{\star\top} \left( \frac{1}{1+\sigma_t^2} \boldsymbol{U}_l^\star \boldsymbol{U}_l^{\star\top} + \frac{\delta^2}{\delta^2+\sigma_t^2} \widetilde{\boldsymbol{U}}_l^\star \widetilde{\boldsymbol{U}}_l^{\star\top} \right) \boldsymbol{x}_t$$

$$= \hat{w}_k^\star \left( \frac{1}{1+\sigma_t^2} \boldsymbol{U}_k^\star \boldsymbol{U}_k^{\star\top} \boldsymbol{x}_t \right) + \sum_{l \neq k} \hat{w}_l^\star \left( \frac{\delta^2}{\delta^2+\sigma_t^2} \boldsymbol{U}_k^\star \boldsymbol{U}_k^{\star\top} \boldsymbol{x}_t \right)$$

$$= \left( \frac{\hat{w}_k^\star}{1+\sigma_t^2} + \frac{(K-1)\delta^2 \hat{w}_l^\star}{\delta^2+\sigma_t^2} \right) \boldsymbol{U}_k^\star \boldsymbol{U}_k^{\star\top} (\boldsymbol{U}_k^\star \boldsymbol{a}_i + \delta \widetilde{\boldsymbol{U}}_k^\star \boldsymbol{e}_i + \sigma_t \boldsymbol{\epsilon}_i)$$

$$= \left( \frac{\hat{w}_k^\star}{1+\sigma_t^2} + \frac{(K-1)\delta^2 \hat{w}_l^\star}{\delta^2+\sigma_t^2} \right) (\boldsymbol{U}_k^\star \boldsymbol{a}_i + \sigma_t \boldsymbol{U}_k^\star \boldsymbol{U}_k^{\star\top} \boldsymbol{\epsilon}_i)$$

Since $\boldsymbol{U}_k \in \mathcal{O}^{n \times d}$:

$$\mathbb{E}[\|\boldsymbol{U}_k^\star \boldsymbol{U}_k^{\star\top} \hat{\boldsymbol{x}}_{approx}^\star(\boldsymbol{x}_t, t)\|^2] = \left( \frac{\hat{w}_k}{1+\sigma_t^2} + \frac{(K-1)\delta^2 \hat{w}_l}{\delta^2+\sigma_t^2} \right)^2 \left( d + \sigma_t^2 d \right),$$

and similarly for (13):

$$\boldsymbol{U}_l^\star \boldsymbol{U}_l^{\star\top} \hat{\boldsymbol{x}}_{approx}^\star(\boldsymbol{x}_t, t) = \hat{w}_k^\star \boldsymbol{U}_l^\star \boldsymbol{U}_l^{\star\top} \left( \frac{1}{1+\sigma_t^2} \boldsymbol{U}_k^\star \boldsymbol{U}_k^{\star\top} + \frac{\delta^2}{\delta^2+\sigma_t^2} \widetilde{\boldsymbol{U}}_k \widetilde{\boldsymbol{U}}_k^{\star T} \right) \boldsymbol{x}_t$$

$$+ \hat{w}_l^\star \boldsymbol{U}_l^\star \boldsymbol{U}_l^{\star\top} \left( \frac{1}{1+\sigma_t^2} \boldsymbol{U}_l^\star \boldsymbol{U}_l^{\star\top} + \frac{\delta^2}{\delta^2+\sigma_t^2} \widetilde{\boldsymbol{U}}_l \widetilde{\boldsymbol{U}}_l^{\star\top} \right) \boldsymbol{x}_t$$

$$+ \sum_{j \neq k,l} \hat{w}_j^\star \boldsymbol{U}_l^\star \boldsymbol{U}_l^{\star\top} \left( \frac{1}{1+\sigma_t^2} \boldsymbol{U}_j^\star \boldsymbol{U}_j^{\star\top} + \frac{\delta^2}{\delta^2+\sigma_t^2} \widetilde{\boldsymbol{U}}_j \widetilde{\boldsymbol{U}}_j^{\star\top} \right) \boldsymbol{x}_t$$

$$= \hat{w}_k^\star \left( \frac{\delta^2}{\delta^2+\sigma_t^2} \boldsymbol{U}_l^\star \boldsymbol{U}_l^{\star\top} \boldsymbol{x}_t \right) + \hat{w}_l^\star \left( \frac{1}{1+\sigma_t^2} \boldsymbol{U}_l^\star \boldsymbol{U}_l^{\star\top} \boldsymbol{x}_t \right) + \sum_{j \neq k,l} \hat{w}_j^\star \left( \frac{\delta^2}{\delta^2+\sigma_t^2} \boldsymbol{U}_l^\star \boldsymbol{U}_l^{\star\top} \boldsymbol{x}_t \right)$$

$$= \left( \frac{\hat{w}_l^\star}{1+\sigma_t^2} + \frac{\delta^2(\hat{w}_k^\star + (K-2)\hat{w}_j^\star)}{\delta^2+\sigma_t^2} \right) \boldsymbol{U}_l^\star \boldsymbol{U}_l^{\star\top} (\boldsymbol{U}_k^\star \boldsymbol{a}_i + \delta \widetilde{\boldsymbol{U}}_k^\star \boldsymbol{e}_i + \sigma_t \boldsymbol{\epsilon}_i)$$

$$= \left( \frac{\hat{w}_l^\star}{1+\sigma_t^2} + \frac{\delta^2(\hat{w}_k^\star + (K-2)\hat{w}_l^\star)}{\delta^2+\sigma_t^2} \right) \left( \delta \boldsymbol{U}_l^\star \boldsymbol{e}_i[1:d] + \sigma_t \boldsymbol{U}_l^\star \boldsymbol{U}_l^{\star\top} \boldsymbol{\epsilon}_i \right)$$

where the third equality follows since $\hat{w}_j^\star = \hat{w}_l^\star$ for all $j \neq k, l$. Further, we have:

$$\mathbb{E}[\|\boldsymbol{U}_l^\star \boldsymbol{U}_l^{\star\top} \hat{\boldsymbol{x}}_{approx}^\star(\boldsymbol{x}_t, t)\|^2] = \left( \frac{\hat{w}_l^\star}{1+\sigma_t^2} + \frac{\delta^2(\hat{w}_k^\star + (K-2)\hat{w}_l^\star)}{\delta^2+\sigma_t^2} \right)^2 \left( \delta^2 d + \sigma_t^2 d \right).$$

Lastly, we prove (14). Given that the subspaces of all classes and the complement space are both orthonormal and mutually orthogonal, we can write:

$$\mathbb{E}[\|\hat{\boldsymbol{x}}_{approx}^\star(\boldsymbol{x}_t, t)\|^2] = \mathbb{E}[\|\boldsymbol{U}_k^\star \boldsymbol{U}_k^{\star\top} \hat{\boldsymbol{x}}_{approx}^\star(\boldsymbol{x}_t, t)\|^2] + \mathbb{E}[\sum_{l \neq k} \|\boldsymbol{U}_l^\star \boldsymbol{U}_l^{\star\top} \hat{\boldsymbol{x}}_{approx}^\star(\boldsymbol{x}_t, t)\|^2] + \mathbb{E}[\|\boldsymbol{U}_\perp \boldsymbol{U}_\perp^T \hat{\boldsymbol{x}}_{approx}^\star(\boldsymbol{x}_t, t)\|^2]$$

where we define the noise space as $\boldsymbol{U}_\perp = \bigcap_{k=1}^K \boldsymbol{U}_k^{\star\perp} \in \mathcal{O}^{n \times (n-Kd)}$, representing the directions orthogonal to all class subspaces. Since $\boldsymbol{U}_\perp$ is orthogonal to each $\boldsymbol{U}_l^\star$, the third term vanishes. Combining the remaining terms, we obtain:

$$\mathbb{E}[\|\hat{\boldsymbol{x}}_{approx}^\star(\boldsymbol{x}_t, t)\|^2] = \left( \frac{\hat{w}_k^\star}{1+\sigma_t^2} + \frac{(K-1)\delta^2 \hat{w}_l^\star}{\delta^2+\sigma_t^2} \right)^2 \left( d + \sigma_t^2 d \right)$$

$$+ (K-1) \left( \frac{\hat{w}_l^\star}{1+\sigma_t^2} + \frac{\delta^2(\hat{w}_k^\star + (K-2)\hat{w}_l^\star)}{\delta^2+\sigma_t^2} \right)^2 \left( \delta^2 d + \sigma_t^2 d \right).$$

$\square$

