# OpenReview forum: "Understanding Representation Dynamics of Diffusion Models via Low-Dimensional Modeling"
_NeurIPS.cc/2025/Conference — NeurIPS 2025 poster_

### Official Review · Reviewer_DRKN · 2025-06-10

**Clarity:** 3
**Significance:** 2
**Originality:** 3
**Rating:** 5
**Confidence:** 4

**Summary:**

The paper offers an interesting theoretical and experimental analysis on the dynamics of latent representations in generative diffusion models. The theoretical analysis solely considers the case of mixtures of gaussians with zero mean and different covariance structure. The analysis is performed by considering a simplified network architecture that can be optimized exactly in the mixture of Gaussians.

The quantity of interest is a form of feature SNR that quantifies the ratio between the norm of the projection of the score on the true subspace with the norm of the projection on its orthogonal complement (noise space). The authors argue that the presence of a single mode in this feature SNR curve is an index of generalization. This particular claim is unfortunately only supported by experimental data and it is not proven theoretically.

It is interesting to understand why the SNR curve has a unimodal shape in the mixture of gaussians cases considered by the authors. The intuition is that the score function s_t(x) at time t is a (approximately) local average over the data points in a ball of radius sigma(t). For large noise levels, the ball is so large to include all components of the mixture, leading to a low feature SNR. On the other hand, for very small noise levels the average is very local and the particles are very close to the target distribution. In this case, for a datapoint close to the shared mean the covariance structure is averaged over the covariance structures of all components, which leads to a somewhat paradoxical reduction of feature SNR.
However, the feature SNR would have a second peak at t = 0 if the different components are allowed to have different means since, in this more general case, the score at the ‘typical points’ will have the same covariance structure of the component-specific score when sigma(t) is small enough to properly separate the mean vectors (assuming that the distance between the mean vectors is larger than the sds of the individual components).
While the authors do not make this point, this can provide a connection with memorization since in the memorization case each datapoint becomes the center of its own Gaussian component [1,2,3,4].

[Edit: I increased my score based on the response to my and other reviews.]

**Questions:**

1) My intuition is that you should see a second peak of the SNR at t = 0 if the centers of the gaussians are well separated. Am I correct? Can you elaborate on this point?

2) Is your network parameterization optimized by the exact mixture of gaussians score? This score can be obtained in closed form and it can be directly analyzed. If not, can you elaborate on the differences and similarities between the constrained optimum and the exact score?

3) Can you connect the peak in the SNR with the speciation/symmetry breaking time identified in [1] and [5]? It would also be useful to discuss the connection with the transitions observed in [6].

Minor observations:
I do not know what an informal theorem is. It would be better to either express Theorem 1 in a more rigourous way or to not present it in theorem format.

In line 133, you claim that the latent space of diffusion models is approximately Gaussian. This is just not true, if it were the case, we would not need fancy diffusion models and we could just sample by fitting the covariance matrix. In reality, the autoencoder has a very mild 'gaussianization' effect even thought it uses a KL regularization term.

**Ethical Concerns:**

["NO or VERY MINOR ethics concerns only"]

**Final Justification:**

The author addressed some important points concerning clarity, connection with pruor literature and assumptions. I think that this work provides a novel view on the theory of generative diffusion that provides insights and complements existing approaches.

**Limitations:**

The discussion of limitations and future directions is rather limited. I would have loved to see a more substantial discussion of the effect of changing their distributional assumptions away from the mixture of gaussians case.

**Paper Formatting Concerns:**

Nothing to report

**Quality:**

3

**Strengths And Weaknesses:**

Generally speaking, I do think that this paper contains good ideas and interesting theoretical and experimental analysis.  The link between the feature SNR and generalization/memorization is insightful and seems to be well supported by the experiments on real images. The theoretical part is somewhat limited but it contains interesting results.

A weakness is the lack of theoretical results connecting the SNR with the memorization phenomenon. A second related weakness is the very limited coverage of very relavant existing literature on generalization, memorization, manifolds and class speciation [1-8]. It is important for the authors to reference and discuss these papers and I believe that the methods used in [1,2,4] could be very useful to connect this analysis to the theory of memorization, which is the main concern of the authors in this paper.

References:

Memorization
[1] Biroli, Giulio, et al. "Dynamical regimes of diffusion models." Nature Communications 15.1 (2024): 9957.
[2] Ambrogioni, Luca. "In search of dispersed memories: Generative diffusion models are associative memory networks." Entropy 26.5 (2024): 381.
[3] Pham, Bao, et al. "Memorization to generalization: Emergence of diffusion models from associative memory." arXiv preprint arXiv:2505.21777 (2025).
[4] Lucibello, Carlo, and Marc Mézard. "Exponential capacity of dense associative memories." Physical Review Letters 132.7 (2024): 077301.

Class speciation:
[1] Biroli, Giulio, et al. "Dynamical regimes of diffusion models." Nature Communications 15.1 (2024): 9957.
[5] Raya, Gabriel, and Luca Ambrogioni. "Spontaneous symmetry breaking in generative diffusion models." Advances in Neural Information Processing Systems 36 (2023): 66377-66389.
[6] Sclocchi, Antonio, Alessandro Favero, and Matthieu Wyart. "A phase transition in diffusion models reveals the hierarchical nature of data." Proceedings of the National Academy of Sciences 122.1 (2025): e2408799121.

Generalization on manifolds:
[7] George, Anand Jerry, Rodrigo Veiga, and Nicolas Macris. "Analysis of Diffusion Models for Manifold Data." arXiv preprint arXiv:2502.04339 (2025).
[8] Achilli, Beatrice, et al. "Memorization and Generalization in Generative Diffusion under the Manifold Hypothesis." arXiv preprint arXiv:2502.09578 (2025).
[3] Achilli, Beatrice, et al. "Losing dimensions: Geometric memorization in generative diffusion." arXiv preprint arXiv:2410.08727 (2024).

---

> ### Author Rebuttal · Authors · 2025-07-31
>
> We thank the reviewer for the thoughtful and detailed feedback. We are glad to hear that the reviewer found our theoretical and experimental analyses interesting, and appreciated the insights linking representation learning to generalization and memorization. Below, we respond to the reviewer’s comments in detail.
>
> > A weakness is the lack of theoretical results connecting the SNR with the memorization phenomenon. A second related weakness is the very limited coverage of very relavant existing literature on generalization, memorization, manifolds and class speciation [1-8].
>
> **Answer**: We are grateful to the reviewer for pointing us to this body of work, and will include a thorough discussion of all the papers above in the updated version of our manuscript. We have carefully reviewed the references provided and summarize their main contributions as follows:
>
>     [1] Studies the dynamics of diffusion model (DM) sampling using an empirical score. The authors define speciation (when the denoised sample falls into one of the data classes) and collapse (when it converges to a discrete training sample).
>     [2] Draws a connection between empirical DMs and dense associative memory (AM) networks through their objectives and sampling behavior.
>     [3] Identifies spurious samples that appear as a DM transitions from memorization to generalization, from an AM perspective.
>     [4] Analyzes the capacity and retrieval behavior of AMs for Gaussian and spherical patterns.
>     [5] Investigates the symmetry breaking in DM sampling, as in [1], and proposes a sampling strategy that starts from a smaller noise scale.
>     [6] Uses the RHM model to show that different levels of features are recovered at different timesteps during DM sampling.
>     [7] Extends the analysis in [1] to data lying on a low-dimensional manifold defined by a linear subspace and an activation function.
>     [8] Analyzes the discrepancy between the empirical and true score during sampling, illustrating how memorization leads to the loss of local geometric components.
>
> We note that [7] and [8] in particular adopt data assumptions closely aligned with ours. We also appreciate the reviewer’s suggestion to relate our SNR analysis to the speciation and symmetry breaking behavior observed in [1] and [5], as well as to the phase transitions analyzed in [6]. We will expand on these connections in the revised manuscript.
>
> > Can you connect the peak in the SNR with the speciation/symmetry breaking time identified in [1] and [5]? It would also be useful to discuss the connection with the transitions observed in [6].
>
> **Answer**:
> In [1,5], the authors characterize the emergence of speciation by tracking the probability that a denoised sample aligns with a class cluster, mostly using gaussian mixtures with well-separated means. Our approach differs in that we model speciation through alignment to subspaces corresponding to different classes, under a low-dimensional data structure. The SNR we define reflects the signal strength along these subspaces, which can be interpreted as a subspace-based analog of speciation. Furthermore, the class confidence metric introduced in our Section 3.4 may be viewed as a probabilistic surrogate analogous to the metrics proposed in [1].
>
> Regarding [6], their analysis reveals that features emerge hierarchically across timesteps, starting from coarse to fine-grained structures. This matches our own observations in Section 3.4, where we note a fine-to-coarse shift on the model output caused by the representations learned by DMs during denoising. However, our focus is on internal representations rather than generated outputs.
>
> >It is important for the authors to reference and discuss these papers and I believe that the methods used in [1,2,4] could be very useful to connect this analysis to the theory of memorization, which is the main concern of the authors in this paper.
>
> **Answer**: We thank the reviewer for this constructive suggestion. We agree that the energy-based perspective in [1, 2, 4] is highly relevant, as these papers offer a powerful description of memorization by analyzing the dynamics of the empirical score, leading to phenomena like 'class speciation.' Our work provides a representation learning perspective on a similar problem, where we extensively show that generalizing DMs effectively learns the low-dimensional data structure and how this learning breaks down when data memorization occurs.
>
> References:
>
>
> [1] Biroli, Giulio, et al. "Dynamical regimes of diffusion models." Nature Communications 15.1 (2024): 9957.
>
> [2] Ambrogioni, Luca. "In search of dispersed memories: Generative diffusion models are associative memory networks." Entropy 26.5 (2024): 381.
>
> [3] Pham, Bao, et al. "Memorization to generalization: Emergence of diffusion models from associative memory." arXiv preprint arXiv:2505.21777 (2025).
>
> [4] Lucibello, Carlo, and Marc Mézard. "Exponential capacity of dense associative memories." Physical Review Letters 132.7 (2024): 077301.
>
> [5] Raya, Gabriel, and Luca Ambrogioni. "Spontaneous symmetry breaking in generative diffusion models." Advances in Neural Information Processing Systems 36 (2023): 66377-66389.
>
> [6] Sclocchi, Antonio, Alessandro Favero, and Matthieu Wyart. "A phase transition in diffusion models reveals the hierarchical nature of data." Proceedings of the National Academy of Sciences 122.1 (2025): e2408799121.
>
> [7] George, Anand Jerry, Rodrigo Veiga, and Nicolas Macris. "Analysis of Diffusion Models for Manifold Data." arXiv preprint arXiv:2502.04339 (2025).
>
> [8] Achilli, Beatrice, et al. "Memorization and Generalization in Generative Diffusion under the Manifold Hypothesis." arXiv preprint arXiv:2502.09578 (2025).
>
> > My intuition is that you should see a second peak of the SNR at t = 0 if the centers of the gaussians are well separated. Am I correct? Can you elaborate on this point?
>
> **Answer**:  We thank the reviewer for raising this insightful point. When the Gaussian class centers are well separated, both classification accuracy and SNR tend to decrease monotonically with increasing noise, the unimodal dynamic disappears.
>
> In this setting, the data remains linearly separable at low noise levels. As $\sigma_t$ increases, the added noise begins to blur the class boundaries, reducing both accuracy and SNR. Thus, instead of a unimodal shape, we observe a monotonic decline.
>
> To validate this, we conduct an experiment with well-separated class means. The results confirm this behavior:
>
> | $\sigma_t$ | 0.030 | 0.053 | 0.098 | 0.189 | 0.282 | 0.379 | 0.480 | 0.588 | 0.704 | 0.830 | 0.989 | 1.492 | 1.717 | 1.978 |
> | :--- | :--- | :--- | :--- | :--- | :--- | :--- | :--- | :--- | :--- | :--- | :--- | :--- | :--- | :--- |
> | **Acc** | 100 | 100 | 100 | 99.8 | 97.7 | 92.4 | 85.7 | 78.5 | 71.8 | 66.0 | 61.6 | 52.6 | 50.1 | 48.0 |
> | **SNR** | 0.540 | 0.539 | 0.535 | 0.524 | 0.513 | 0.502 | 0.484 | 0.465 | 0.436 | 0.406 | 0.370 | 0.297 | 0.266 | 0.240 |
>
> > Is your network parameterization optimized by the exact mixture of gaussians score? This score can be obtained in closed form and it can be directly analyzed. If not, can you elaborate on the differences and similarities between the constrained optimum and the exact score?
>
> **Answer**: Thanks for the good catch. Our parameterization is indeed inspired by the closed-form posterior mean of the MoLRG model, since a denoising auto-encoder should estimate that mean. Nonetheless, we note that our network is not fitted by directly plugging in the closed-form MoLRG score; instead, it is trained with the standard denoising objective (Eq. 2).
> In essence, our network acts as a constrained function approximator for the true score. The architectural constraints enforce the correct functional form, while the standard denoising training finds the parameters of that function that best fit the data. The success of this approach is validated by Figure 3, where the SNR of our trained network closely tracks the theoretical optimum. We will make this connection more explicit in the updated manuscript.
>
> > Minor observations: I do not know what an informal theorem is. It would be better to either express Theorem 1 in a more rigorous way or to not present it in theorem format.
>
> **Answer**: Thanks for pointing this out. We will follow the reviewer's suggestion to rename “Theorem 1 (Informal)” in the main text to “Key result (informal)” and add a pointer to the formal version of the Theorem in Appendix E. We hope this can prevent any confusion and maintain rigor.
>
> > In line 133, you claim that the latent space of diffusion models is approximately Gaussian. This is just not true, if it were the case, we would not need fancy diffusion models and we could just sample by fitting the covariance matrix. In reality, the autoencoder has a very mild 'gaussianization' effect even thought it uses a KL regularization term.
>
> **Answer**: Thanks for pointing out this problem. We will soften the wording from “The latent space of latent diffusion models is approximately Gaussian” to “Latent diffusion models encourage the latent space toward a Gaussian distribution”.

---

> > ### Comment · Reviewer_DRKN · 2025-08-03
> >
> > Thank you for the thorough reply. I am happy with the response and I think that the modifications will substantially improve the manuscript. I also appreciated the response to the other reviewers concerning the assumption of orthogonality, which is something that I did not notice in my original review. In general, I think that the euthors did a great job in their response.
> >
> > I will therefore increase my score and recommend acceptance.

---

> > > ### Author Response · Authors · 2025-08-03
> > >
> > > We appreciate the reviewer’s positive reply and kind words. We are grateful for the suggestions raised and will ensure they are incorporated into the revised manuscript. Thank you again for the thoughtful review and feedbacks.

---

### Official Review · Reviewer_9v41 · 2025-07-02

**Clarity:** 3
**Significance:** 3
**Originality:** 4
**Rating:** 5
**Confidence:** 4

**Summary:**

This paper analyzes the representations of diffusion models from two perspectives.

First, it theoretically explains why *unimodal representation dynamics*—the phenomenon where the quality of learned features peaks at intermediate diffusion timesteps—arises. Specifically, the paper shows the following:

* It assumes the data is a low-dimensional mixture of Gaussian distributions (MoLRG), and the model is a very simple linear U-Net combined with a mixture-of-experts-like architecture.
* Under this setting, it derives the model’s optimal parameters (Proposition 1).
* It defines the SNR (Definition 1) as a metric to measure the quality of the model’s representations for classification and provides an analytic expression for SNR (Theorem 1).
* The paper shows that this analytical SNR matches both empirical SNR and feature accuracy in synthetic experiments (Figure 3) and also explains trends on real datasets like CIFAR-10 and TinyImageNet (Figure 5).

Second, the paper empirically investigates the relationship between diffusion model representations and generalization. For example:

* It shows that unimodal dynamics are linked to generalization as measured by feature accuracy: when unimodal dynamics are absent, feature accuracy is very low.
* It demonstrates that feature accuracy correlates with the memorization behavior of generative models: the trend of high FID – low FID – high FID (underfitting – generalization – memorization) observed during training iterations also appears when measuring feature accuracy.

Overall, this is a strong paper that provides an analytic understanding of representation learning in diffusion models and connects it empirically to memorization, one of the interesting phenomena in generative modeling.

**Questions:**

* I assume feature accuracy was measured using a linear classifier with logits defined as the inner product between the class embedding vector and the feature vector. Have you considered trying an L2 similarity-based classifier? Since the SNR is defined via the L2 norm, I wonder if this might reveal even more consistent trends, although I understand that training might be more challenging.

**Ethical Concerns:**

["NO or VERY MINOR ethics concerns only"]

**Final Justification:**

I think this paper successfully builds a theoretical framework to explain the representation learning of the diffusion model. There are some downsides, such as (1) logical flow of the paper (2) over simplified setting, but I believe overall this paper contributes significantly to the theory of understanding the diffusion model.

**Limitations:**

* The paper does a good job describing limitations, such as the fact that the analysis tools, assumptions, and scope of experiments (e.g., synthetic data models and classification-focused benchmarks) may not capture the full complexity of all applications.

**Quality:**

4

**Strengths And Weaknesses:**

**Strengths**

* Despite being theoretically heavy, the paper is very easy to understand. The writing is clear, and the visualizations are excellent.
* To my knowledge, this is the first work to provide an analytic treatment of diffusion model representations. This is a very impressive contribution. Beyond simply describing trends, the paper demonstrates that the derived model can accurately predict the peak in unimodal dynamics of feature accuracy (Figures 3 and 5), which is particularly impressive.

**Weaknesses**

* This is not a major issue, but the connection between the first part (theory) and the second part (generalization and memorization) is not very tight. Fundamentally, this may be hard to address, as the theoretical analysis assumes an almost-linear model. Linear models, by their nature, have limited expressivity and regularization effects, which makes them less appropriate for analyzing the generalization–memorization behavior observed in neural networks. It would be nice if the first and second parts were linked a bit more naturally.

---

> ### Author Rebuttal · Authors · 2025-07-31
>
> We thank the reviewer for the detailed and generous review. We are pleased to hear that the reviewer finds our theoretical analysis of diffusion model representations clear, novel, and impactful, and appreciates our empirical visualizations. Below, we address the reviewer’s comments in detail.
>
> > This is not a major issue, but the connection between the first part (theory) and the second part (generalization and memorization) is not very tight. Fundamentally, this may be hard to address, as the theoretical analysis assumes an almost-linear model. Linear models, by their nature, have limited expressivity and regularization effects, which makes them less appropriate for analyzing the generalization–memorization behavior observed in neural networks. It would be nice if the first and second parts were linked a bit more naturally.
>
> **Answer**: Thank you for the thoughtful feedback. We agree that the current theoretical analysis does not directly cover the generalization–memorization transition described in the latter part of the paper.
>
> Our main theoretical contribution is to show that unimodal representation dynamics emerge when the diffusion model effectively captures the low-dimensional data distribution. The second part of our study aims to reinforce this view by demonstrating that, under limited data, this distribution learning fails, leading to a breakdown of the unimodal behavior.
>
> As the reviewer noted, a complete theoretical understanding of this transition will likely require analyzing a more nonlinear model. In particular, such a model would need to capture both the fitting of the true data distribution when data is abundant, and the overfitting to the empirical distribution when data is limited. Interestingly, our experiments suggest that diffusion models first align with the true distribution before adapting to the empirical one (as discussed in Section 4.2), hence a learning dynamic based analysis would also be interesting.
>
> We are actively investigating this phenomenon as a future work and would greatly welcome any further insights or suggestions from the reviewer.
>
> > I assume feature accuracy was measured using a linear classifier with logits defined as the inner product between the class embedding vector and the feature vector. Have you considered trying an L2 similarity-based classifier? Since the SNR is defined via the L2 norm, I wonder if this might reveal even more consistent trends, although I understand that training might be more challenging.
>
> **Answer**: Thanks for the insightful suggestion. Yes, our linear probing experiments use logistic regression classifiers with inner-product-based logits.
>
> Inspired by the reviewer's comment, we re-do the experiments in Figure 5 using a projection-based classifier: for each class, we compute its principal subspace $[V_1, ..., V_K]$ from the training features, and classify each test sample by identifying the class whose subspace captures the most projection energy, computed as $\text{argmax}_k ||h(x_i) V_k||^2$. The results are shown below. While the projection-based classifier yields lower overall accuracy, it indeed reveals a more pronounced unimodal trend that aligns closely with the SNR dynamics. We will discuss these results in the updated manuscript.
>
> | Dataset       | Classifier Type                  | $\sigma_t = 0.002$ | $0.008$ | $0.023$ | $0.060$ | $0.140$ | $0.296$ | $0.585$ | $1.088$ | $1.923$ | $3.257$ |
> |:-------------:|:--------------------------------:|:------------------:|:-------:|:-------:|:-------:|:-------:|:-------:|:-------:|:-------:|:-------:|:-------:|
> | CIFAR10       | Logistic Regression (Figure 5)   | 91.32              | 93.72   | 94.94   | 95.20   | 94.11   | 91.36   | 85.25   | 72.95   | 56.21   | 41.98   |
> | CIFAR10       | Projection                       | 78.93              | 86.95   | 90.36   | 90.93   | 90.78   | 88.48   | 82.82   | 70.53   | 53.84   | 37.84   |
>
> | Dataset       | Classifier Type                  | $\sigma_t = 0.002$ | $0.008$ | $0.023$ | $0.060$ | $0.140$ | $0.296$ | $0.585$ | $1.088$ | $1.923$ | $3.257$ |
> |:-------------:|:--------------------------------:|:------------------:|:-------:|:-------:|:-------:|:-------:|:-------:|:-------:|:-------:|:-------:|:-------:|
> | TinyImageNet  | Logistic Regression (Figure 5)   | 31.13              | 32.17   | 34.70   | 43.09   | 53.58   | 50.78   | 42.13   | 27.56   | 15.10   | 8.39    |
> | TinyImageNet  | Projection                       | 12.02              | 12.89   | 14.38   | 24.10   | 34.78   | 34.88   | 30.02   | 20.99   | 12.64   | 7.89    |

---

> > ### Comment · Reviewer_9v41 · 2025-08-05
> >
> > I have considered this paper to be making a very important theoretical contribution from the beginning, and after reading the rebuttal and the other reviewers' comments, I see no issue with it being accepted. Thank you for the excellent work. I will maintain my score as Accept.

---

> > > ### Author Response · Authors · 2025-08-07
> > >
> > > We thank the reviewer for the positive evaluation and kind words. We will incorporate the suggested experiments into the revised manuscript, as well as pursue the future directions suggested by the reviewer. Thank you again for the valuable insights and feedback.

---

### Official Review · Reviewer_95ka · 2025-07-03

**Clarity:** 2
**Significance:** 3
**Originality:** 3
**Rating:** 4
**Confidence:** 3

**Summary:**

This paper develops a mathematical framework to analyze the expressiveness of features learned by denoising diffusion probabilistic models (DDPMs), with a focus on how this expressiveness varies across timesteps. The analysis is grounded in an assumed data distribution, a noisy low-rank Gaussian mixture, and a parametrization of the denoising network resembling eigen-decomposition. The theoretical findings align with empirical observations: features extracted at intermediate timesteps tend to be more expressive, as measured by downstream task performance. Additional experiments further demonstrate that this behavior correlates with the generalizability of the diffusion model.

**Questions:**

- Could you clarify and justify the data distribution assumptions used in the theory?
- How is the proposed network parameterization connected to U-Net architectures used in practice, and how does this impact the validity of the theoretical conclusions?

**Ethical Concerns:**

["NO or VERY MINOR ethics concerns only"]

**Final Justification:**

Please see my response to the authors' rebuttal.

**Limitations:**

I do not see any major ethical or societal concerns raised by this work. However, a more explicit discussion of the limitations and the realism of the theoretical assumptions would be helpful for readers and could strengthen the paper.

**Paper Formatting Concerns:**

I did not notice any.

**Quality:**

3

**Strengths And Weaknesses:**

**Strengths**:
- To the best of my knowledge, this is the first work that provides a theoretical analysis of the relationship between the feature expressiveness of diffusion models and the denoising timestep.
- Under the stated assumptions, the derivation offers a sound theoretical explanation for the empirical phenomenon under investigation.
- The experiments reveal interesting and potentially useful connections between feature expressiveness and practical aspects of diffusion model training.
- The experimental evaluation is extensive, with validation across multiple model variants, datasets, and evaluation metrics.

**Weaknesses**:
- Several critical assumptions underlying the theoretical framework are insufficiently justified. The authors should consider discussing these assumptions further or citing prior works that support them.
  - While the data distribution assumption is discussed to some extent, others remain unaddressed. For instance, is it reasonable to assume that data from different classes lie in orthogonal subspaces? Similarly, is the modeling of class-independent fine-grained details as belonging to subspaces of other classes theoretically or empirically supported?
  - The parameterization of DAE is underexplained. It is claimed to resemble a shallow U-Net, but this connection is not clearly established. Furthermore, most diffusion models in practice use deeper U-Nets. The feature extractor is modeled as a linear projection followed by a self-attention-like reweighting, which differs from standard U-Net architectures. Additionally, the feature is defined in the theory as a linear projection from the input data itself, whereas in the experiments (as described in the supplementary material), features are extracted from intermediate layers of the network. These intermediate features are not obviously related to a direct linear projection from the data space, raising questions about how well the theoretical definition aligns with the practical implementation.
- Proposition 1 assumes that the latent feature dimensionality equals the intrinsic dimensionality of the data. This assumption is difficult to justify, especially given that deep networks are often overparameterized and operate in much higher-dimensional latent spaces. The authors should either justify this assumption more explicitly or discuss how relaxing it might affect the conclusions.
- There is a notational inconsistency regarding the DAE. It is denoted as $x_\theta$ in Section 3.2, but as $\hat{x}_\theta$ in Definition 1, without explanation. Clarifying this would improve readability.

---

> ### Author Rebuttal · Authors · 2025-07-31
>
> We thank the reviewer for the thoughtful and detailed review. We are glad to see that the reviewer recognizes the novelty and contributions of our theoretical analysis, as well as the extensive experimental validation. We also appreciate the reviewer's recognition of the connection we draw between feature learning and generalization in diffusion model training. Below, we address the reviewer’s comments point by point.
>
> >While the data distribution assumption is discussed to some extent, others remain unaddressed. For instance, is it reasonable to assume that data from different classes lie in orthogonal subspaces? Similarly, is the modeling of class-independent fine-grained details as belonging to subspaces of other classes theoretically or empirically supported?
>
> **Answer**:  We thank the reviewer for raising this important point. Below we address the reviewer’s concerns:
>
> **The assumption that different classes lie in othogonal subspaces.** This assumption is primarily made to simplify the analysis and enable closed-form derivations. However, this is not essential for the validity of our main findings. In practice, subspaces corresponding to different classes may not be perfectly orthogonal, but as long as the principal angles between them are sufficiently large, our theoretical results remain valid with appropriate technical refinements.
>
> To support our claim beyond this assumption, we conduct an ablation study where we control the principal angle $\theta$ between class subspaces. As shown below, the unimodal SNR trend remains present even with substantial subspace overlap, suggesting that it is not sensitive to the orthogonality assumption.
>
> | Training Setting                           | SNR @  $\sigma_t = 0.002$ | $0.008$ | $0.023$ | $0.060$ | $0.140$ | $0.296$ | $0.585$ | $1.088$ | $1.923$ | $3.257$ |
> |:-----------------------------------------:|:------------------:|:-----:|:-----:|:-----:|:-----:|:-----:|:-----:|:-----:|:-----:|:-----:|
> | $\theta = 90^{\circ}$ (Non-overlapping)   |       24.85        | 24.88 | 25.15 | 26.95 | 35.84 | 58.27 | 20.84 |  4.05 |  1.57 |  1.13 |
> | $\theta = 30^{\circ}$ (Overlapping)       |       16.67        | 16.70 | 16.92 | 18.39 | 25.97 | 38.54 | 22.88 | 14.56 | 12.55 | 12.13 |
>
> **The modeling of class-independent fine-grained details as belonging to subspaces of other classes.**  Our data model assumes that the dataset lies in a union of subspaces, where each class is associated with a dominant, class-specific subspace, and there exists a shared subspace common across all classes that captures class-independent fine-grained details. This type of shared–specific decomposition has been empirically supported in the literature on subspace clustering and representation learning (e.g., [1,2]).
>
> We will include these discussions, along with supporting ablation results, in the revised manuscript to clarify and validate this modeling assumption.
>
> [1] Bousmalis et al., Domain Separation Networks, NeurIPS 2016
>
> [2] Zhou et al., Dual Shared-Specific Multiview Subspace Clustering, IEEE Transactions on Cybernetics, 2019
>
> >The parameterization of DAE is underexplained. It is claimed to resemble a shallow U-Net, but this connection is not clearly established. Furthermore, most diffusion models in practice use deeper U-Nets. The feature extractor is modeled as a linear projection followed by a self-attention-like reweighting, which differs from standard U-Net architectures. Additionally, the feature is defined in the theory as a linear projection from the input data itself, whereas in the experiments (as described in the supplementary material), features are extracted from intermediate layers of the network. These intermediate features are not obviously related to a direct linear projection from the data space, raising questions about how well the theoretical definition aligns with the practical implementation. How is the proposed network parameterization connected to U-Net architectures used in practice, and how does this impact the validity of the theoretical conclusions?
>
> **Answer**:  Our primary goal with the simplified model was to isolate the core mechanism behind unimodal representation dynamics. While the parameterization does not fully match a modern U-Net, it is sufficient to reproduce this key behavior on both synthetic data (Figure 3) and real datasets.
> To validate this, we train our simplified DAE on the full CIFAR10 dataset and report the SNR curve below. The unimodal trend persists, suggesting that this behavior is not an artifact of complex functionalities within a modern U-Net, but rather rooted in the intrinsic structure of the data, and hence can be captured with a simple model like ours.
> | Network                          | Dataset |  $\sigma_t = 0.002$ | $0.008$ | $0.023$ | $0.060$ | $0.140$ | $0.296$ | $0.585$ | $1.088$ | $1.923$ | $3.257$ |
> | :--------------------: | :-------------------: | :---: | :---: | :---: | :---: | :----: | :----: | :----: | :---: | :---: | :---: |
> | Simplifed DAE  | CIFAR10 |         1.39 |   1.39 |   1.41 |   1.49 |  1.88 | 2.54 | 2.78 | 2.63 |    2.20 |    1.33 |   0.36 |
>
> That said, we conjecture that the more complex architecture of full U-Nets may influence feature evolution and shift the peak location. We agree with the reviewer that a more comprehensive theoretical treatment of U-Nets remains an important direction for future work.
>
> >Proposition 1 assumes that the latent feature dimensionality equals the intrinsic dimensionality of the data. This assumption is difficult to justify, especially given that deep networks are often overparameterized and operate in much higher-dimensional latent spaces. The authors should either justify this assumption more explicitly or discuss how relaxing it might affect the conclusions.
>
> **Answer**: We thank the reviewer for this valuable comment. The equal-dimensionality setting ($d' = d$) for feature dimension ($d'$) and data intrinsic dimension ($d$) is used only to keep the theoretical results clean and to isolate the essential factors behind the unimodal representation dynamics. The result extends naturally to an overcomplete feature space. Specifically, for any $d' > d$, we may take $U_{l,\text{new}}^{\star}=\begin{bmatrix} U_{l}^{\star} & 0_{d'-d} \end{bmatrix} \in \mathbb{R}^{n\times d'}$ and any permutation or orthogonal rotation applied within each class block of this form would make the loss, the soft-max weights and the SNR remain unchanged, and preserve the optimality, although the solution is no longer unique. The conclusions of Proposition 1 remain intact.
>
> We appreciate the reviewer’s attention to this detail and will clarify this extension in the revised manuscript.
>
> > There is a notational inconsistency regarding the DAE. It is denoted as $x_{\theta}$  in Section 3.2, but as $\hat{x_{\theta}}$ in Definition 1, without explanation. Clarifying this would improve readability.
>
> **Answer**: We appreciate the reviewer for raising this concern. We want to clarify that we intend to use $x_{\theta}$ to denote an arbitrary DAE for any parameter set $\theta$ where we use $\hat{x_{\theta}}$ to denote a trained DAE. Because Definition 1 evaluates SNR with the trained network, we use $\hat{x_{\theta}}$ here.
> We will follow the reviewer's suggestion to add a sentence in Section 3.2 that explicitly states this convention and ensure the same notation is used consistently throughout.

---

> > ### Comment · Reviewer_95ka · 2025-08-04
> >
> > I thank the authors for their thorough and thoughtful responses. All of the concerns I raised in my initial review have been satisfactorily addressed. Accordingly, I am raising my score. I would encourage the authors to briefly discuss the potential limitations introduced by abstracting the problem for theoretical convenience, which could help clarify the scope and applicability of the results.

---

> > > ### Author Response · Authors · 2025-08-04
> > >
> > > We appreciate the reviewer’s response and are glad to hear that our rebuttal has addressed the concerns raised. We also thank the reviewer for the helpful suggestion, we will include a more concrete discussion of the assumptions used in the theoretical analysis, as well as the ablations we conducted to explore the effect of relaxing these assumptions in the updated manuscript. Thank you again for the thoughtful review and follow-up.

---

### Official Review · Reviewer_cQRG · 2025-07-08

**Clarity:** 3
**Significance:** 2
**Originality:** 3
**Rating:** 4
**Confidence:** 4

**Summary:**

The paper shows that if images lie on a noisy mixture of low-rank Gaussians and the diffusion denoiser is idealized as a block-diagonal “mixture-of-experts,” then one can derive a closed-form signal-to-noise ratio (SNR) for learned features that rises at low noise, peaks at an intermediate noise level, and then falls, explaining the empirically observed “unimodal” representation quality in diffusion models and validates this theory on both synthetic and real vision benchmarks, also demonstrating that the peak SNR reliably indicates generalization versus memorization and can serve as a principled early-stopping criterion.

**Questions:**

- How well does the Mixture of Low-Rank Gaussians (MoLRG) actually capture the structure of real high-resolution image data (e.g. ImageNet)? What happens if class subspaces overlap or have different ranks?
- To what extent does the block-diagonal “mixture‐of-experts” abstraction preserve the key representational behaviors of full U-Net architectures with attention and skip‐connections? Could cross‐block interactions materially change the unimodality result?
- The derivation of the approximate SNR curve hinges on swapping the expectation and softmax (Eq. 10). Under what conditions might this approximation break down, and how sensitive is the peak location to that error?
- The SNR metric is evaluated for classification accuracy. Would an analogous unimodal behavior hold for other downstream tasks, e.g. segmentation, detection, or self‐supervised objectives?
- The paper suggests that losing unimodality signals onset of memorization. How robust is this indicator across dataset sizes, architectures, and training regimes?
- Do similar unimodal representation dynamics appear in latent diffusion models (e.g. Stable Diffusion) where the denoiser operates in a lower‐dimensional embedding space?

**Ethical Concerns:**

["NO or VERY MINOR ethics concerns only"]

**Final Justification:**

The authors properly addressed most of my concerns. I'll raise the score to 4 as promised.

**Limitations:**

I'm happy to raise my score if my concerns are addressed properly.

**Quality:**

3

**Strengths And Weaknesses:**

Strengths:
- The assumption that images lie on a noisy mixture of low-rank Gaussians (MoLRG) is supported by a growing body of empirical work on intrinsic dimensionality in vision data. It lets the authors derive closed-form expressions for the optimal denoiser and an explicit SNR metric (Eq. 6) that quantifies feature quality.
- By modeling the denoising autoencoder as a block-diagonal “mixture-of-experts” layer (Eq. 4–5), they capture key behavior of U-Net–style architectures while keeping the math clean. This lets them prove (Prop. 1) that the optimal basis aligns with the true subspaces and derive an analytical form for the denoiser.
-  The informal Theorem 1 shows the SNR follows a unimodal curve because as noise $\sigma_t$ increases, a “denoising rate” term grows while a “class confidence” term decays, yielding a single peak in their ratio (Eq. 8).

Weaknesses:
- Real data rarely satisfy exact orthonormal subspaces or uniform mixing weights. The analysis assumes all subspaces have the same dimension and are mutually orthogonal—conditions that hold only approximately in practice.
- Actual diffusion models use deep, highly nonlinear U-Nets with skip connections, attention, and non-diagonal feature mixing. Reducing them to a single block-diagonal layer may miss interactions that affect real-world representation dynamics.
- The analysis treats a decaying constant $C_t$ as negligible (“minimal impact to unimodality”), but in practice its behavior across $t$ could shift the peak. Furthermore, the metric is derived for classification accuracy—its relevance to other downstream tasks (segmentation, alignment) may require additional work.
- Some missing related works should be added in the discussion [1, 2, 3]. There might be others but I suggest the authors examine the literature carefully.

[1] Infodiffusion: Representation learning using information maximizing diffusion models

[2] Lossy image compression with conditional diffusion models

[3] Soda: Bottleneck diffusion models for representation learning

---

> ### Author Rebuttal · Authors · 2025-07-31
>
> We thank the reviewer for the thoughtful and constructive feedback. We're glad the MoLRG assumption and theoretical results were appreciated. Below, we address the reviewer’s concerns one by one.
>
> > Real data rarely satisfy exact orthonormal subspaces or uniform mixing weights. The analysis assumes all subspaces have the same dimension and are mutually orthogonal—conditions that hold only approximately in practice. What happens if class subspaces overlap or have different ranks?
>
> **Answer**: We appreciate this important point. As the first theoretical study on diffusion-based unimodal representations, we adopt a simplified setting to isolate the root cause of unimodality. That said, we agree it is valuable to test the robustness of our findings under relaxed assumptions. Empirically, we observe that the unimodal SNR trend persists even when class subspaces overlap, differ in rank, or follow non-uniform mixing:
>
> 1. Overlapping Class Subspaces: We control the principal angle ($\theta$) between class subspaces and observe that overlap tend to reduce SNR across timesteps while the peak remains stable. We conjecture that overlap can be seen as introducing additional intrinsic noise beyond the $\delta$-related term and thus affect the SNR value.
>
> | Training Setting |SNR @  $\sigma_t = 0.008$|$0.02$|$0.06$|$0.14$ | $0.30$ | $0.59$ | $1.09$ | $1.92$ |
> |:------:|:-----:|:-----:|:-----:|:-----:|:-----:|:-----:|:-----:|:-----:|
> | $\theta = 90^{\circ}$ (Non-overlapping)| 24.9|25.2|27.0|35.8|58.3|20.8|4.1|1.6|
> | $\theta = 30^{\circ}$ (Overlapping) |16.7|16.9|18.4|26.0|38.5|22.9|14.6|12.6|
>
> 2. Varying Subspace Ranks: We set class subspace dimensions to $d_0=10, d_1=2$. Intuitively, the higher-rank class retains more signal and is less sensitive to noise, yielding higher and later-peaking SNR. The low-rank class decays earlier.
>
> | Training Setting | SNR @  $\sigma_t = 0.008$ | $0.02$ | $0.06$ | $0.14$ | $0.30$ | $0.59$ | $1.09$ | $1.92$ |
> |:------:|:-----:|:-----:|:-----:|:-----:|:-----:|:-----:|:-----:|:-----:|
> | Class 0 ($d_0 = 10$) | 69.0 | 69.8 | 74.9 | 102.2 | 195.4 | 215.3 | 34.2 | 8.4 |
> | Class 1 ($d_1 = 2$) | 7.8 |  7.8 |  8.1 |  8.8  |  6.5  |  1.8 |  0.5 |  0.3|
> | Overall SNR | 24.1| 24.3 | 25.2 | 28.2 | 23.3|  8.2 | 2.8| 1.5 |
>
> 3. Non-Uniform Mixing Weights: We set $\pi_0=0.8, \pi_1=0.2$ and observe consistently higher SNR for the majority class. We conjecture that this may stem from both the score function of the distribution and DAE training being biased toward denoising more frequent samples.
>
> | Training Setting | SNR @  $\sigma_t = 0.008$ | $0.02$ | $0.06$ | $0.14$ | $0.30$ | $0.59$ | $1.09$ | $1.92$|
> |:------:|:-----:|:-----:|:-----:|:-----:|:-----:|:-----:|:-----:|:-----:|
> | Class 0 ($\pi_0 = 0.8$) |52.4 | 53.0 | 56.7|75.6|134.8|72.4|13.6| 4.5|
> |  Class 1 ($\pi_1 = 0.2$) |12.2 |12.3|13.1|17.5|23.9 |6.1|1.2|0.5|
> | Overall SNR| 38.7| 39.2|41.9|55.8 |90.5 |35.3|7.4| 2.8|
>
> In summary, these new results demonstrate that the unimodal SNR behavior generalizes to more complex and realistic data distributions beyond our theoretical setup. We will include this expanded study in the revision.
>
> > How well does the Mixture of Low-Rank Gaussians (MoLRG) actually capture the structure of real high-resolution image data (e.g. ImageNet)?
>
> **Answer**: We thank the reviewer for this question. As noted in Section 3.1, high-resolution datasets like ImageNet exhibit low intrinsic dimensionality [1], which our MoLRG is designed to model.
>
> Moreover, prior work [2] used MoLRG to simulate real data and produced qualitatively reasonable samples resembling those from pre-trained diffusion models. While no analytical model fully captures high-resolution image complexity, MoLRG offers a reasonable abstraction of the key low-dimensional structure.
>
> [1] Pope et al; The Intrinsic Dimension of Images and Its Impact on Learning; ICLR 2021.
>
> [2] Wang et al; Diffusion Models Learn Low-Dimensional Distributions via Subspace Clustering; 2024.
>
> >Actual diffusion models use deep, highly nonlinear U-Nets with skip connections, attention, and non-diagonal feature mixing. Reducing them to a single block-diagonal layer may miss interactions that affect real-world representation dynamics.
>
> **Answer**:  We agree with the reviewer that U-Net architectures are more complex, but this also makes theoretical analysis being intractable. Our simplified model aims to isolate the core mechanisms behind unimodal representation dynamics under the MoLRG assumption, which we find sufficient for explaining key unimodal representation behaviors.
>
> Additionally, many prior works (see for example [3–5]) have similarly adopted simplified architectures when modeling Gaussian-like data, enabling theoretical insights into otherwise complex systems. A more complete theoretical treatment of U-Nets is an important future direction, and we view our analysis as a first step in that direction.
>
> [3] Shah et al; Learning mixtures of Gaussians using the DDPM objective; NeurIPS 2024.
>
> [4] Ji et al; The Power of Contrast for Feature Learning: A Theoretical Analysis; JMLR.
>
> [5] Han et al; On the Feature Learning in Diffusion Models; ICLR 2025.
>
> > To what extent does the block-diagonal “mixture‐of-experts” abstraction preserve the key representational behaviors of full U-Net architectures with attention and skip‐connections? Could cross‐block interactions materially change the unimodality result?
>
> **Answer**: Despite its simplified block-diagonal structure, our DAE reproduces the key unimodal SNR trend observed in full diffusion models when trained on CIFAR10. While complex U-Net components may influence feature evolution or shift the peak, the core dynamic is captured by our simplified model.
>
> |Network|Dataset|$\sigma_t = 0.008$|$0.02$|$0.06$|$0.14$|$0.30$|$0.59$|$1.09$|$1.92$|$3.26$|
> | :-----: | :---: | :---: | :---: | :---: | :----: | :----: | :----: | :---: | :---: | :---: |
> | Simplifed DAE|CIFAR10|1.39|1.41|1.49|1.88|2.54|2.78|2.63|2.20|1.33|0.36|
>
> >The analysis treats a decaying constant $C_t$ as negligible (“minimal impact to unimodality”), but in practice its behavior across t could shift the peak.
>
> **Answer**: In light of the reviewer's suggestion, we ablate the effect of $C_t$ in Theorem 1 under the MoLRG setting and find that while it alters the SNR magnitude, it has minimal impact on the peak location.
>
> | Data Setting|SNR Type| $\sigma_t = 0.008$|$0.02$|$0.06$|$0.14$|$0.30$|$0.59$|$1.09$|$1.92$|
> | :-----: | :---: | :---: | :---: | :---: | :----: | :----: | :----: | :---: | :---:|
> | $n=50,d=10,K=3,\delta=0.1$|w/ $C_t$|50.3| 52.6 |67.7|144.5|449.9|1310.9|30.6|1.1|0.6|
> |  $n=50,d=10,K=3,\delta=0.1$|w/o $C_t$|0.5|0.6|0.9|4.2|40.5|344.3|16.7|0.9|0.5|
> | $n=50,d=2,K=10,\delta=0.2$|w/ $C_t$|2.78|2.81|3.02|4.05|8.00|2.35|0.30|0.15|0.12|
> |  $n=50,d=2,K=10,\delta=0.2$|w/o $C_t$|0.11|0.12|0.13|0.24|0.94|0.67|0.17|0.12|0.11|
>
> >Some missing related works should be added in the discussion [1, 2, 3]. There might be others but I suggest the authors examine the literature carefully.
>
> **Answer**: We thank the reviewer for these relevant references. Due to space constraints, we postponed the related work section (including [1] and [3]) to Appendix A. Following the reviewer’s suggestions, we incorporated [2] and other recent works on diffusion-based representations. These changes will be updated in the manuscript.
>
> >The derivation of the approximate SNR curve hinges on swapping the expectation and softmax (Eq. 10). Under what conditions might this approximation break down, and how sensitive is the peak location to that error?
>
> **Answer**: We appreciate the reviewer’s careful attention to the derivation. To assess the robustness of the approximation, we varied $K$, $d$, and $\delta$ and found the peak may shift slightly, but the unimodal SNR trend remains stable (Figure 14, Appendix C).
>
> >The SNR metric is evaluated for classification accuracy. Would an analogous unimodal behavior hold for other downstream tasks, e.g. segmentation, detection, or self‐supervised objectives? the metric is derived for classification accuracy—its relevance to other downstream tasks (segmentation, alignment) may require additional work.
>
> **Answer**: We agree that our current theory is framed in a classification setting via subspace-aligned SNR, and extending it to other tasks may require task-specific SNR formulations which is a promising direction. We believe that the core SNR principle: trading off signal preservation and noise suppression should be applicable broadly. Empirically, unimodal trends also appear in segmentation (Figure 1b) and correspondence [6,7], suggesting the phenomenon generalizes beyond classification.
>
> [6] Baranchuk et al; Label-Efficient Semantic Segmentation with Diffusion Models; 2022.
>
> [7] Tang et al; Emergent Correspondence from Image Diffusion; 2023.
>
> >The paper suggests that losing unimodality signals onset of memorization. How robust is this indicator across dataset sizes, architectures, and training regimes?
>
> **Answer**: The indicator is quite robust. In addition to the results in the main paper, we include experiments on subsets of Oxford-IIIT Pet (3680 images) and TinyImageNet (2048 images) in Figures 9 and 10 of the Appendix. In both, loss of unimodal dynamics signals the onset of memorization.
>
> > Do similar unimodal representation dynamics appear in latent diffusion models (e.g. Stable Diffusion) where the denoiser operates in a lower‐dimensional embedding space?
>
> **Answer**: Yes, unimodal dynamics persist beyond pixel-space denoisers. Using DiT-XL/2 on miniImageNet, we observe similar trends in feature accuracy and SNR, confirming the effect holds in latent diffusion models.
>
> | $\sigma_t$ |0.03|0.04|0.08|0.11| 0.14| 0.21|0.27|0.34|0.72|1.23|2.03|
> | :--- | :--- | :--- | :--- | :--- | :--- | :--- | :--- | :--- | :--- | :--- | :--- |
> | Acc|74.6|75.2|75.5|75.8|75.9|76.1|75.9|74.9|66.5|49.3|29.3|
> | SNR |0.0372|0.0375|0.0380|0.0382|0.0384|0.0387|0.0384|0.0387|0.0350|0.0321|0.0282|

---

### Note · Authors · 2025-08-14

Dear AC and Reviewers,

We sincerely appreciate your valuable comments and constructive discussions during the rebuttal. We are encouraged by the positive feedback from the reviewers. As Reviewer **95ka** noted: *“All of the concerns I raised in my initial review have been satisfactorily addressed. Accordingly, I am raising my score.”* Reviewer **DRKN** also stated: *“The authors did a great job in their response. I will therefore increase my score and recommend acceptance.”* Reviewer **cQRG** highlighted that our theory *“captures the key behavior of UNet model architectures”*, and Reviewer **9v41** maintained a highly positive stance: *“Thank you for the excellent work. I will maintain my score as Accept.”*

We **addressed the main concerns** raised by the reviewers during the rebuttal as follows:

- *Validity of our data and model assumptions.* We conducted extensive ablation experiments to show that our proposed metric continues to capture the unimodal trend even when these assumptions are relaxed. These assumptions are used primarily for analytical simplicity.
- *Related work.* Reviewers provided valuable pointers to related studies on representation learning in diffusion models, as well as on phase transitions and class speciation during sampling. We have examined these works in detail and identified meaningful connections to our study.
- *Additional experiments.* We added experiments with different settings, such as alternative probing methods, model architectures, and dataset sizes, confirming that the unimodal trend persists across these variations.

**For paper revision**, besides addressing the points above, we will also conduct the following:

- Include additional ablations on the MoLRG data assumptions and the network assumption, such as covering overlapping subspaces, varying subspace ranks, different mixing weights, and well-separated means in Appendix D.
- Add further experiments, including L2-similarity–based probing results and unimodal results for latent diffusion models in Appendix B.
- Incorporate the related works suggested by Reviewers **DRKN** and **cQRG**, with further discussion on the connections to prior work and potential limitations of our approach.

Overall, we remain confident in the contributions of this work and are grateful to the AC and reviewers for helping us strengthen its depth and impact.

---

### Decision · Program_Chairs · 2025-09-17

**Decision:**

Accept (poster)

**Comment:**

This paper develops a low-dimensional analytical account of unimodal representation dynamics in diffusion models and tests its implications empirically. Under a mixture-of-low-rank-Gaussians data model and a simplified denoiser, the authors derive an explicit metric for representation quality that rises with denoising strength at small noise and falls with waning class confidence at large noise, yielding a single peak. They show that this metric predicts the empirical peak in feature quality across timesteps on synthetic data and standard vision benchmarks, and they further observe that the presence of a unimodal curve tracks generalization, while its erosion accompanies memorization. Reviewers found the problem important and viewed the analysis as a first theoretical treatment of diffusion-model representations; they also noted practical value in using the peak as an empirical indicator (e.g., for early stopping).

Strengths include (i) a clear theoretical mechanism explaining unimodality that aligns with measurements, (ii) a unifying SNR formalism that connects denoising and class confidence, and (iii) broad empirical support across datasets, probing schemes, and architectures. The main weaknesses are the reliance on stylized assumptions (orthogonal class subspaces, matched ranks, simplified denoiser), gaps in formally tying SNR to memorization, and some missing or deferred related work and notation/clarity issues. During rebuttal the authors ran additional ablations (overlapping subspaces, unequal ranks, non-uniform mixing), added alternative probing that strengthened the SNR link, and showed the trend in latent diffusion models; they clarified approximations and notation, softened claims, and committed to incorporating suggested literature. All reviewers vote for acceptance. I recommend acceptance.